# Role of the V2R–βarrestin–Gβγ complex in promoting G protein translocation to endosomes
Badr Sokrat [1,2,5,6], Anthony H. Nguyen [3,6], Alex R. B. Thomsen [3,4,5], Li-Yin Huang [3], Hiroyuki Kobayashi [2], Alem W. Kahsai [4], Jihee Kim [3], Bing X. Ho [3], Symon Ma [3], John Little IV [3], Catherine Ehrhart [3], Ian Pyne [3], Emmery Hammond [3] & Michel Bouvier [1,2] ✉

Classically, G protein-coupled receptors (GPCRs) promote signaling at the plasma membrane through activation of heterotrimeric Gαβγ proteins, followed by the recruitment of GPCR kinases and βarrestin (βarr) to initiate receptor desensitization and internalization. However, studies demonstrated that some GPCRs continue to signal from internalized compartments, with distinct cellular responses. Both βarr and Gβγ contribute to such non-canonical endosomal G protein signaling, but their specific roles and contributions remain poorly understood. Here, we demonstrate that the vasopressin $V_2$ receptor ($V_2$R)–βarr complex scaffolds Gβγ at the plasma membrane through a direct interaction with βarr, enabling its transport to endosomes. Gβγ subsequently potentiates $Gα_s$ endosomal translocation, presumably to regenerate an endosomal pool of heterotrimeric $G_s$. This work shines light on the mechanism underlying G protein subunits translocation from the plasma membrane to the endosomes and provides a basis for understanding the role of βarr in mediating sustained G protein signaling.

G protein-coupled receptors (GPCRs) are the largest class of membrane receptors encoded by the human genome and are involved in the regulation of virtually every physiological process[1,2]. These receptors share a common hepta-helical transmembrane structure and are activated by a large variety of extracellular stimuli, including small molecules, hormones, neuro-transmitters, lipids, and peptides[3,4]. Upon binding to an agonist at its orthosteric binding site accessible via the extracellular compartment, GPCRs adopt active conformations that enable the intracellular engagement of heterotrimeric Gαβγ proteins by the receptor at the plasma membrane[2]. This engagement catalyzes the exchange of GDP for GTP in the Gα subunit, leading to the dissociation of the Gβγ heterodimer from the Gα subunit[2]. The GTP-bound Gα subsequently interacts with effectors such as adenylyl cyclase to generate second messengers like cyclic AMP (cAMP) in order to propagate a wave of signaling that eventually results in a cellular response[1,5,6].

To prevent over-activation of these signaling pathways, GPCRs undergo a desensitization process mediated by receptor phosphorylation and βarrestin (βarr) recruitment. This is initiated by the phosphorylation of a GPCR at specific serine and threonine residues located within the intra-cellular cytoplasmic loops (ICLs) and/or C-terminal tail of the receptor by

GPCR kinases (GRKs)[7]. The phosphorylation increases the affinity of the receptor for βarr enabling its recruitment, thus sterically hindering G pro-tein coupling to the receptor[8,9]. Notably, we previously demonstrated that GPCR–βarr complexes can adopt two distinct conformations: (1) whereby βarr engages the phosphorylated tail of the receptor (deemed the "tail" conformation) or (2) whereby βarr additionally engages the intracellular core of the GPCR via its finger loop region (deemed the "core" conformation)[9]. Additionally, we and others demonstrated that a GPCR–βarr complex in the tail conformation can carry out most functions expected of a receptor-activated βarr with the exception of desensitization, which is exclusively carried out by the core conformation[10–12]. βarr also contributes to several signaling events through scaffolding a variety of other enzymes, such as mitogen activated protein kinases and ubiquitin ligases[13–17].

βarr also recruits endocytic proteins such as the adaptor protein complex (AP2) and clathrin to facilitate receptor internalization into early endosomes, where some receptors rapidly lose their interaction with βarr (known as class A GPCRs) whereas others maintain a sustained interaction with βarr (known as class B GPCRs)[18–20]. Several class B GPCRs such as the

[1]Department of Biochemistry and Molecular Medicine, University of Montreal, Montreal, QC H3T 1J4, Canada. [2]Institute for Research in Immunology and Cancer, University of Montreal, Montreal, QC H3T 1J4, Canada. [3]Department of Biochemistry, Duke University School of Medicine, Durham, NC 27710, USA. [4]Department of Medicine, Duke University Medical Center, Durham, NC 27710, USA. [5]Present address: Department of Molecular Pathobiology, New York University School of Dentistry, New York, NY 10010, USA. [6]These authors contributed equally: Badr Sokrat, Anthony H. Nguyen. ✉e-mail: michel.bouvier@umontreal.ca

parathyroid hormone receptor (PTHR), neurokinin 1 receptor ($NK_1R$) and the vasopressin type 2 receptor ($V_2R$) have been shown to continue signaling within internalized compartments instead of remaining desensitized[21–23]. Initially, this mode of sustained signaling has been difficult to integrate into the classical model of signaling, which states that βarr sterically hinders additional G protein coupling at receptors. However, additional investigations by us and others showed that sustained signaling may involve the formation of a GPCR–βarr–$G_s$ megacomplex in endosomes. This "megaplex" comprises a βarr which engages the receptor in a tail conformation, thus leaving the receptor intracellular core free to couple to and activate a heterotrimeric G protein within endosomes[24,25]. The megaplex provides a potential biophysical explanation for how certain GPCRs continues to signal within internalized compartments.

While βarr classically serves as a desensitizer of G protein signaling at the plasma membrane, it can serve to potentiate sustained G protein signaling from within internalized compartments. Interestingly, βarr1 has also been shown to interact with Gβγ to promote Akt phosphorylation and NF-κB activation[26,27]. Additional reports demonstrate that Gβγ significantly influences sustained G protein signaling by the PTHR, a prototypical class B GPCR, potentially through the formation of a PTHR–βarr–Gβγ complex[21,28,29]. Five distinct G protein β subunits and twelve G protein γ subunits have been identified, which can pair to form distinct heterodimeric Gβγ combinations[30]. Several studies have shown that specific Gβγ heterodimers are found in different intracellular membranes such as the Golgi, ER, mitochondria and endosomes[31–33]. Also, although $G_s$ is anchored at the plasma membrane via palmitoylation, previous works have demonstrated that receptor activation leads to $G_s$ dissociation from the plasma membrane to the cytoplasm[34–38]. This de-palmitoylation reaction is mediated by enzymes such as acyl-protein thioesterases[38] and the cytoplasmic pool of $G_s$ has been shown to translocate to various subcellular compartments[39].

Collectively, these observations raise several questions: (1) despite its classical role in receptor desensitization, how does βarr enhance sustained G protein signaling, particularly at class B GPCRs? (2) Within the GPCR–βarr–Gβγ complex, what is the role of Gβγ in mediating said signaling? To answer these questions, we employed a variety of cellular, biochemical and biophysical approaches to elucidate the mechanism of endosomal trafficking of $G_s$ and Gβγ using the $V_2R$ as a prototypical class B GPCR.

## Results
### $G_s$ dissociates from the plasma membrane after $V_2R$ activation and translocates to endosomes

To investigate $G_s$ trafficking from the plasma membrane to the endosomal compartment, we monitored $G_s$ translocation using an enhanced bystander bioluminescence resonance energy transfer (ebBRET) approach[40]. First, we assessed $G_s$ translocation from the plasma membrane to the endosomes in HEK293T cells as a function of time following $V_2R$ stimulation with arginine-vasopressin (AVP). This was done by measuring the signal between BRET donor $G_s$67-RlucII and BRET acceptor *Renilla reniformis* GFP (rGFP) anchored at the plasma membrane (Supplementary Fig. 1a) via a prenylated CAAX motif or the endosomes (Supplementary Fig. 1b) using the FYVE targeting domain of endofin[41]. A stimulation time of 20 minutes was then chosen as it corresponds to the maximal signals (Supplementary Fig. 1). As expected, $V_2R$ activation caused a decrease in BRET signal at the plasma membrane, indicating dissociation of Gαs from the plasma membrane into the cytosol (Fig. 1a), consistent with previous observations[34,39,42]. Interestingly, expression of a plasma membrane anchored GRK2 C-terminal peptide (referred to as βARKct-CAAX) that acts as an inhibitory scavenger of Gβγ[43,44] caused a significant decrease in $G_s$ dissociation from the plasma membrane. In contrast, overexpression of $Gβ_1$ and $Gγ_2$ subunits (from this point forward referred to as Gβγ) increased the level of dissociation of $G_s$ from the plasma membrane, this effect being reduced by the addition of βARKct-CAAX. These results suggest that the dissociation of $G_s$ from the plasma membrane is dependent on the presence of free Gβγ and the formation of heterotrimeric $G_s$. The effect of

scavenging free Gβγ with βARKct on $G_s$ release from the plasma membrane was found to be restricted to this compartment since anchoring βARKct to the endosomes using the FYVE targeting domain of endofin[45] had little impact on $G_s$ dissociation from the plasma membrane with or without Gβγ overexpression (Fig. 1b). The potentiating effect of overexpressing Gβγ on Gα dissociation from the plasma membrane is attenuated by overexpression of βARKct in the FYVE containing domain (endosomes) most likely due to the scavenging of Gβγ in this compartment, thus reducing its impact at the plasma membrane. To assess the role of βarr in $G_s$ dissociation from the plasma membrane, we used CRISPR βarr1 and βarr2 knock-out (βarr1/2 KO) HEK293T cells. βarr depletion had no impact on the decrease in BRET between $G_s$67-RlucII and rGFP-CAAX, suggesting that βarr is not required for $G_s$ dissociation from the plasma membrane (Fig. 1a). Similarly, βARKct-CAAX inhibited $G_s$ capacity to leave the plasma membrane in this KO cell line to a similar extent as observed in the parental WT cells (Fig. 1a).

We then investigated the role of Gβγ and βarr in $G_s$ trafficking to early endosomes by measuring the BRET signal between donor $G_s$67-RlucII and acceptor rGFP fused to FYVE[40]. We observed an agonist-promoted increase in BRET signal, indicating accumulation of $G_s$ in endosomes (Fig. 1c). βARKct-CAAX as well as βARKct-FYVE completely blocked $G_s$ trafficking to the endosomes while overexpression of Gβγ enhanced $G_s$ endosomal translocation found to be significant in Fig. 1d and to be a tendency in Fig. 1c, in agreement with the statistically significant increase in dissociation from the plasma membrane (Fig. 1a). Strikingly, we observed a significant reduction in $G_s$ translocation to endosomes in βarr1 and βarr2 KO cells (Fig. 1c), whereas $G_s$ dissociation from the plasma membrane was not impaired by βarr1 and βarr2 depletion (Fig. 1a). Taken together, these results show that sequestration of Gβγ impairs both $G_s$ dissociation from the plasma membrane and endosomal translocation while increasing free Gβγ enhances this $G_s$ trafficking. However, depletion of βarr only impairs $G_s$ endosomal translocation, not its dissociation from the plasma membrane.

### βarr mediates Gβγ trafficking from the plasma membrane to the endosomes

Given that Gα and Gβγ dissociate after receptor activation, as confirmed by the agonist-induced BRET decrease between RlucII-117-Gα and GFP10-Gγ2 indicating dissociation of the Gα and Gβγ subunit, as well as the recruitment of GRK2-GFP10 to the Gβγ-RlucII dimer that requires dissociation of Gα (Supplementary Fig. 2), and that both scavenging of Gβγ and loss of βarr significantly impairs the translocation of $G_s$ to endosomes (Fig. 1c,d), we hypothesized that βarr may be involved in the shuttling of Gβγ from the plasma membrane to endosomes. Endosomal Gβγ could then attract the $G_s$ released from the plasma membrane, allowing the reconstitution of a trimeric G protein in the endosomal compartment. To test this hypothesis, we measured the BRET signal between Gγ2-RlucII and the plasma membrane marker rGFP-CAAX at AVP-stimulated $V_2R$ in both parental and βarr1 and βarr2 KO cells. In the parental cell line, we observed an AVP-induced decrease in BRET at the plasma membrane, reflecting a loss of plasma membrane Gβγ most likely resulting from its internalization. This loss of plasma membrane Gβγ was largely abolished in βarr1 and βarr2 KO cells but restored by transfection of βarr1 and βarr2, suggesting that βarr plays an important role in Gβγ internalization from the plasma membrane (Fig. 1e).

Concomitant with the loss of Gβγ from the plasma membrane, we observed an increase in BRET between Gγ2-RlucII and the endosomal marker rGFP-FYVE, indicating an influx of Gβγ into this compartment (Fig. 1f). Again, this signal was greatly blunted in βarr1 and βarr2 depleted cells while overexpression of βarr1 and βarr2 restored Gβγ trafficking to the endosomes to similar levels to the one observed in parental cells. The weak residual endocytosis signal observed in βarr KO cells is most likely the result of an internalization mechanism that does not require βarr. Taken together, these data suggest that Gβγ undergoes βarr-mediated endocytosis upon $V_2R$ activation.

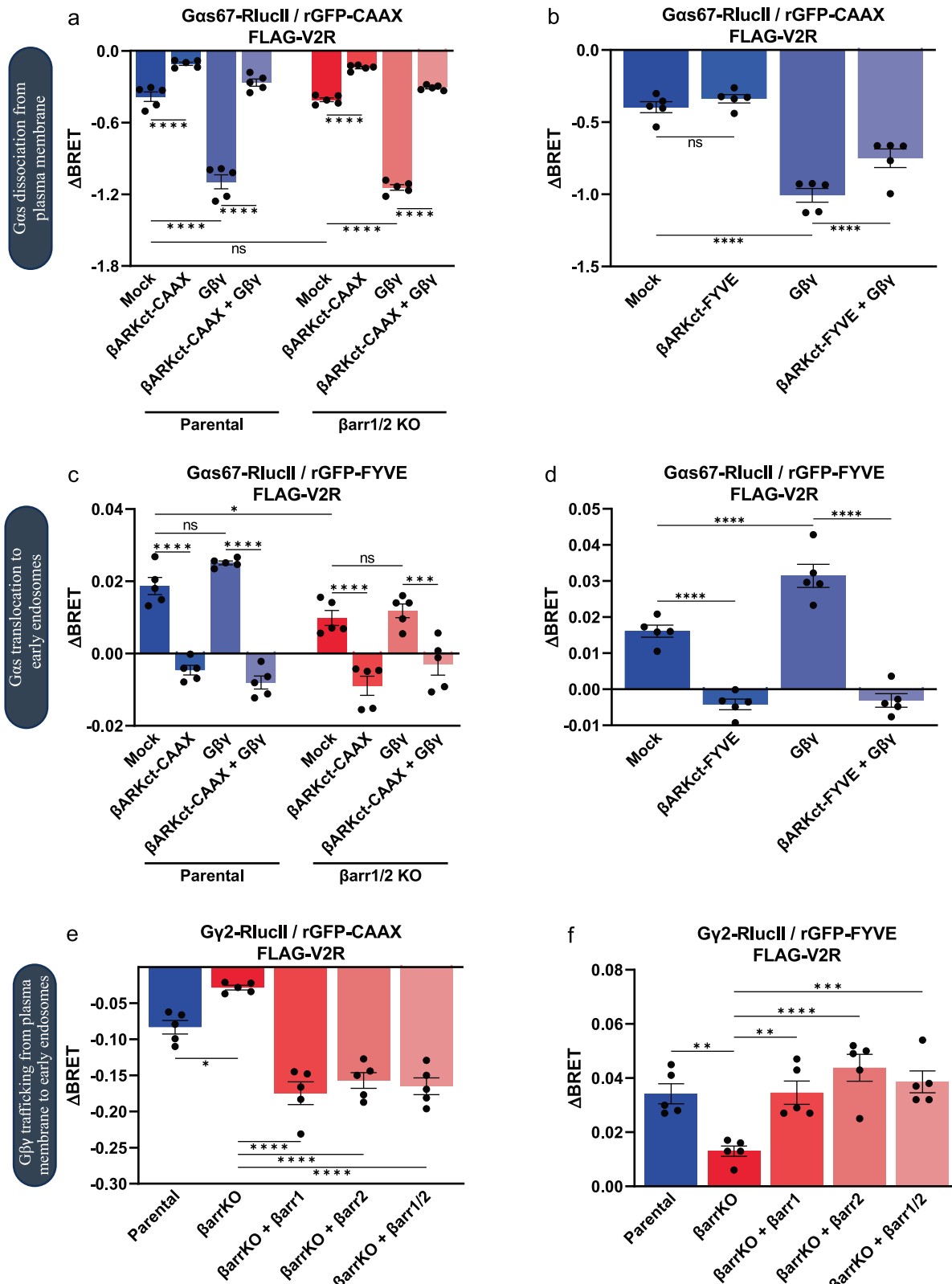

To determine whether the lack of Gβγ trafficking to endosomes in the absence of βarr is merely due to a lack of receptor internalization or a direct βarr-dependent process, we assessed the Gβγ trafficking upon activation of CXCR4, a GPCR that although couples to βarr[46,47], can be internalized via both βarr-dependent and independent pathways. In contrast to what is observed for the V₂R, for which trafficking from the plasma membrane to

the endosomes requires βarr (Fig. 2a, Supplementary Fig. 3a), CXCR4 undergoes agonist-promoted endocytosis in βarr1 and βarr2 KO HEK293T cells (Fig. 2b, Supplementary Fig. 3b) upon CXCL12 stimulation. Despite this βarr-independent internalization of CXCR4, which is comparable in βarr1 and βarr2 KO cells and parental cells, we observed a significant reduction in CXCL12-induced Gβγ dissociation from the plasma

**Fig. 1 | Regulation of G proteins trafficking from the plasma membrane to the endosomes by Gβγ and βarr. a** AVP-induced (100 nM) Gα$_s$ dissociation from the plasma membrane after 20 min stimulation monitored by ebBRET between Gα$_s$67-RlucII and rGFP-CAAX in parental HEK293SL cells and βarr1 and βarr2 KO (βarr1/2 KO) cells. Overexpression of βARKct-CAAX and Gβ$_1$γ$_2$ modulates Gα$_s$ dissociation from the plasma membrane. **b** AVP-induced (100 nM) Gα$_s$ dissociation from the plasma membrane after 20 min stimulation monitored by ebBRET between Gα$_s$67-RlucII and rGFP-CAAX in parental HEK293SL cells. Overexpression of βARKct-FYVE and Gβ$_1$γ$_2$ modulates Gα$_s$ dissociation from the plasma membrane. **c** AVP-induced (100 nM) Gα$_s$ trafficking to the endosomes after 20 min stimulation monitored by ebBRET between Gα$_s$67-RlucII and rGFP-FYVE in parental HEK293SL cells and βarr1/2 KO cells. Overexpression of βARKct-CAAX and Gβ$_1$γ$_2$ modulates Gα$_s$ translocation to early endosomes. **d** AVP-induced (100 nM) Gα$_s$

trafficking to the endosomes after 20 min stimulation monitored by ebBRET between Gα$_s$67-RlucII and rGFP-FYVE in parental HEK293SL cells. Overexpression of βARKct-FYVE and Gβ$_1$γ$_2$ modulates Gα$_s$ translocation to early endosomes. **e** AVP-induced (100 nM) Gβγ internalization after 20 min stimulation monitored by ebBRET between Gγ$_2$-RlucII and rGFP-CAAX in parental HEK293SL cells and βarr1/2 KO cells with or without βarr1 and βarr2 supplementation (βarr1/2 indicating that both βarrs were co-transfected). **f** AVP-induced (100 nM) Gβγ endosomal trafficking after 20 min stimulation monitored by ebBRET between Gγ$_2$-RlucII and rGFP-FYVE in parental HEK293SL cells and βarr1/2 KO cells with or without βarr1 and βarr2 supplementation. Data are represented as the mean ± SEM ($n = 5$) and statistical significance of the differences was assessed using a two-way ANOVA followed by Holm-Šídák's multiple comparison test (ns nonsignificant; *$P \leq 0.05$; **$P \leq 0.01$; ***$P \leq 0.001$; ****$P \leq 0.0001$).

**Fig. 2 | CXCR4-mediated Gβγ trafficking is βarr-dependent. a** V$_2$R internalization is monitored by ebBRET using V$_2$R-RlucII and rGFP-CAAX in parental HEK293SL cells and βarr1 and βarr2 KO (βarr1/2 KO) cells with or without βarr1 and βarr2 supplementation after 20 min 100 nM AVP stimulation. **b** CXCR4 internalization is monitored by ebBRET using CXCR4-RlucII and rGFP-CAAX in parental HEK293SL cells and βarr1/2 KO cells with or without βarr1 and βarr2 supplementation after 20 min 100 nM CXCL12 stimulation. **c** CXCL12-induced (100 nM) Gβγ internalization after 20 min stimulation monitored by ebBRET between Gγ$_2$-RlucII and rGFP-CAAX in parental HEK293SL cells and βarr1/2 KO cells with or without βarr1 and βarr2 supplementation. Data are represented as the mean ± SEM ($n = 4$–5) and statistical significance of the differences was assessed using a two-way ANOVA followed by Holm-Šídák's multiple comparison test (ns nonsignificant; **$P \leq 0.01$; ***$P \leq 0.001$; ****$P \leq 0.0001$).

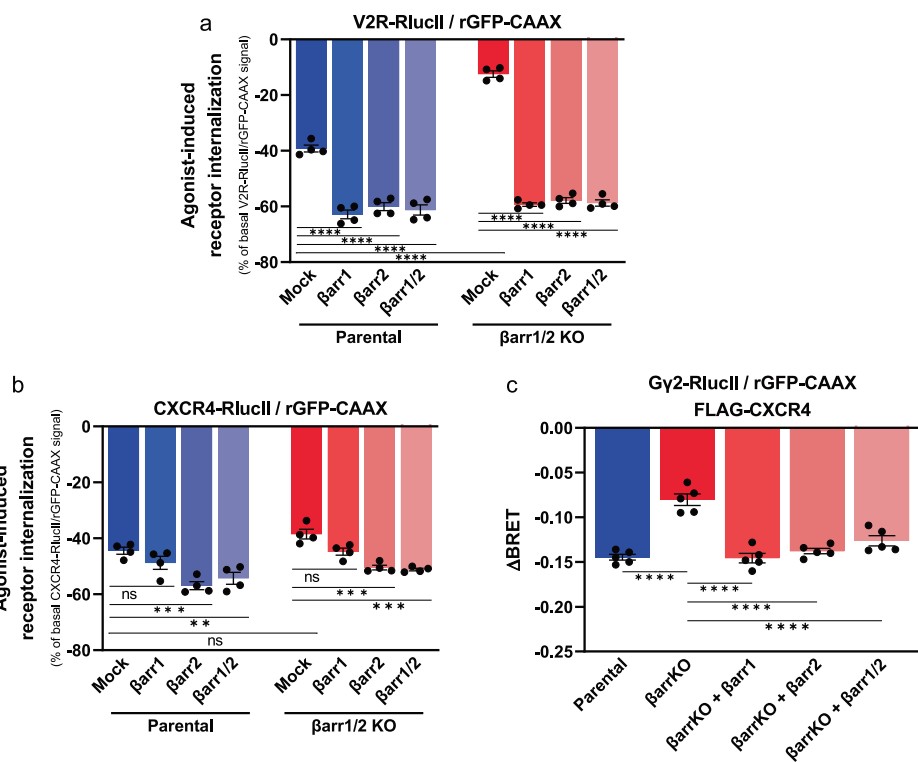

membrane in βarr1 and βarr2 KO cells compared to that in parental cells (Fig. 2c). The blunted Gβγ trafficking was readily rescued with over-expression of either βarr1 or βarr2 (Fig. 2c). These results confirm that Gβγ trafficking from the plasma membrane to the endosomes is βarr-dependent and that receptor internalization is not sufficient to promote Gβγ translocation from the plasma membrane.

## V$_2$R, βarr and Gβγ form a complex in cells

Our previous data suggest that βarr mediates the trafficking of Gβγ from the plasma membrane to the endosomes while Gα$_s$ dissociates from the plasma membrane via a βarr-independent mechanism. Considering that βarr is also essential for V$_2$R internalization, we investigated whether a complex composed of V$_2$R, βarr2, and Gβγ could form in the absence of Gα and be responsible for the endocytosis of Gβγ. To this end, we took advantage of BRET with fluorescence enhancement by combined transfer (BRETfect). This approach tracks the formation of ternary protein complexes by measuring the increase in energy transfer from a luciferase energy donor to a fluorescent energy acceptor in the presence of a fluorescent intermediate[48]. To assess the formation of the complex, we used RlucII fused to βarr2 as an energy donor (D), mTFP fused to the V$_2$R as an energy intermediate (I) and YFP fused to Gγ$_2$ as an energy acceptor (A) (Fig. 3a).

In parental HEK293T cells (Fig. 3b), expression of βarr2-RlucII with V$_2$R-mTFP (D + I) resulted in an AVP-induced increase in signal, indicative of recruitment of βarr2 to the receptor. Expression of βarr2-RlucII with Gγ$_2$-YFP (D + A) did not result in an agonist-induced response in the absence of overexpressed V$_2$R, consistent with the fact that HEK293T cells do not express endogenous V$_2$R. Overexpression of unlabeled V$_2$R with βarr2-RlucII and Gγ$_2$-YFP (D + A) results in a small but robust signal increase after stimulation, indicative of a βarr2-Gβγ interaction after stimulation (Supplementary Fig. 4). However, expression of all three plasmids (D + I + A) produced a significantly higher increase in AVP-induced signal as compared to both βarr2-RlucII/V$_2$R-mTFP (D + I) and βarr2-RlucII/Gγ$_2$-YFP/V$_2$R (D + A) expressing cells, indicating the formation of a ternary complex between V$_2$R, βarr2 and Gβγ (Fig. 3b).

To assess whether the V$_2$R–βarr–Gβγ complex detected by BRETfect can be formed in the absence of Gα subunit, we monitored V$_2$R–βarr2–Gβγ complex formation in Gα$_s$-depleted cells[49], Gα$_s$ being the primary Gα subunit engaged by V$_2$R. We observed a similar agonist-induced BRETfect signal than the one observed in parental cells, indicating the formation of a V$_2$R–βarr2–Gβγ complex in Gα$_s$-depleted cells (Fig. 3c). Since a recent study showed that the V$_2$R also activates Gα$_q$, Gα$_{11}$, Gα$_{13}$, Gα$_{14}$ and Gα$_{15}$[41], we tested complex formation in a cell line lacking all Gα proteins (ΔGNAS/

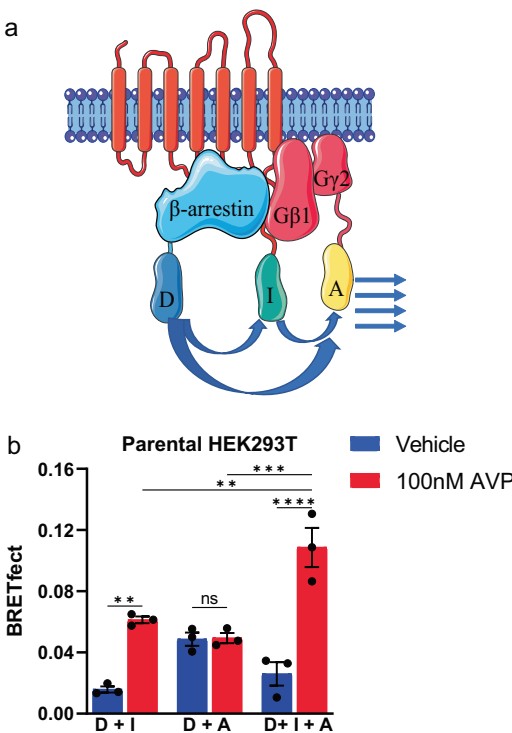

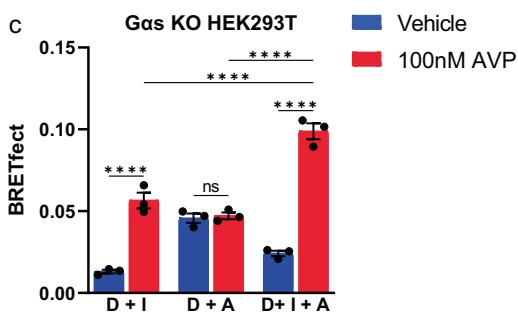

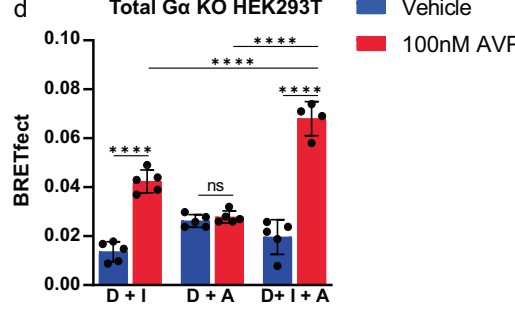

**Fig. 3 | V₂R–βarr–Gβγ complex formation monitored by BRETfect assay.**
**a** Illustration of the design of the BRETfect assay with transfer of energy between RlucII donor (D) fused to βarr2, mTFP intermediate (I) fused to V₂R and energy acceptor YFP (A) fused to Gγ₂. **b** Co-expression of BRETfect constructs in parental HEK293T followed by vehicle or 100 nM AVP stimulation for 20 min. **c** Co-expression of BRETfect constructs in Gα$_s$ KO cells followed by 100 nM AVP stimulation for 20 min. **d** Co-expression of BRETfect constructs in total Gα proteins

KO (ΔGNAS/GNAL/GNAQ/GNA11/GNA12/GNA13/GNAI1/GNAI2/GNAI3/GNAO1/GNAZ/GNAT1/GNAT2) cells followed by 100 nM AVP stimulation for 20 min. Data are represented as the mean ± SEM (n = 3–5) and statistical significance of the differences was assessed using a two-way ANOVA followed by Holm-Šídák's multiple comparison test (ns nonsignificant; **$P \leq 0.01$; ***$P \leq 0.001$; ****$P \leq 0.0001$).

GNAL/GNAQ/GNA11/GNA12/GNA13/GNAI1/GNAI2/GNAI3/GNAO1/GNAZ/GNAT1/GNAT2) to eliminate the possibility of the formation of the previously reported V₂R–βarr–G$_s$ megaplex[24] confounding our BRETfect results. V₂R–βarr2–Gβγ complex could still form in the total absence of Gα proteins although the BRETfect signal was smaller than in the parental cell line (Fig. 3d). Taken together, these results show that V₂R–βarr–Gβγ complex can exist as a unique entity without the incorporation of Gα subunits.

**Agonist-promoted V₂R–βarr2–Gβγ complex formation occurs at the plasma membrane**

The BRETfect approach also allows for real-time imaging of ternary complex formation using BRET microscopy[50]. A moderate agonist-promoted increase in signal could be observed between βarr2-RlucII and V₂R-mTFP at the plasma membrane, reflecting the recruitment of βarr2 to the receptor. As was the case for the spectrometric experiments shown above, the AVP-promoted signal increase observed at the plasma membrane was greatly potentiated in the BRETfect configuration (i.e: co-expression of βarr2-RlucII, V₂R-mTFP and Gγ₂-YFP), supporting the notion that a V₂R–βarr2–Gβγ complex is formed at the plasma membrane (Fig. 4, Supplementary Movie 1).

Kinetic analysis of the BRETfect signal using both imaging and spectrometric approaches revealed that formation of the V₂R–βarr2–Gβγ occurs rapidly after stimulation of the receptor ($t_{1/2}$: 16.7 s) in WT parental HEK293T cells (Fig. 5a–c, Supplementary Fig. 5a). Although the formation of the V₂R–βarr2–Gβγ complex was also observed in Gα$_s$-depleted (Supplementary Fig. 5b) and total Gα KO cells, the formation kinetics was much slower ($t_{1/2}$: 288.8 s) in the absence of Gα subunits (Fig. 5b, c). Taken together, these results show V₂R–βarr2–Gβγ complex formation at the plasma membrane, both in presence and in absence of Gα subunits,

although the former leads to faster complex formation (Fig. 5c, Supplementary Movie 2).

To further assess the potential role of the Gα subunits in the formation of the V₂R–βarr–Gβγ complex, we tested the effect of different Gα subtypes. Whereas over-expression of the Gα subtypes known to be activated by V₂R (i.e.,: Gα$_s$ and Gα$_q$) did not significantly affect complex formation in the parental HEK293T cells, Gα$_{i1}$ and Gα$_{12}$ over-expression resulted in a significant decrease in BRETfect signal, reflecting an inhibition of complex formation (Fig. 6a). In cells lacking all Gα subunits expression (total Gα KO cells), the reintroduction of Gα$_s$ and Gα$_q$ potentiated V₂R–βarr–Gβγ complex formation. In contrast, over-expression of Gα$_{i1}$ and Gα$_{12}$, reduced complex formation maybe by titrating the Gβγ away from the complex and/or preventing βarr engagement (Fig. 6b). This observation is consistent with a previous study that showed unproductive coupling between V₂R and Gα$_{12}$ resulting in an inhibition of agonist-promoted effector recruitment to the receptor and downstream signaling[51]. Another study also showed the formation of a V₂R–βarr–Gα$_i$ complex that does not mediate canonical G protein signaling[52]. Taken together, these data indicate that G protein activation potentiates V₂R–βarr–Gβγ complex formation.

To assess whether V₂R–βarr–Gβγ complex formation occurs at the plasma membrane immediately following receptor activation or may require endocytosis, we investigated the effect of a dominant-negative mutant of dynamin (DynK44A) that inhibits receptor endocytosis[53] and of βARKct peptide that can sequester Gβγ either at the plasma membrane (βARKct-CAAX) or in the endosomes (βARKct-FYVE). DynK44A had no impact on complex formation (Fig. 6c) whereas it blocked V₂R internalization (Supplementary Fig. 6). In contrast, plasma membrane-anchored but not endosomal targeted βARKct drastically blocked complex formation (Fig. 6c, d), indicating that the V₂R–βarr–Gβγ complex forms at the plasma membrane before receptor internalization.

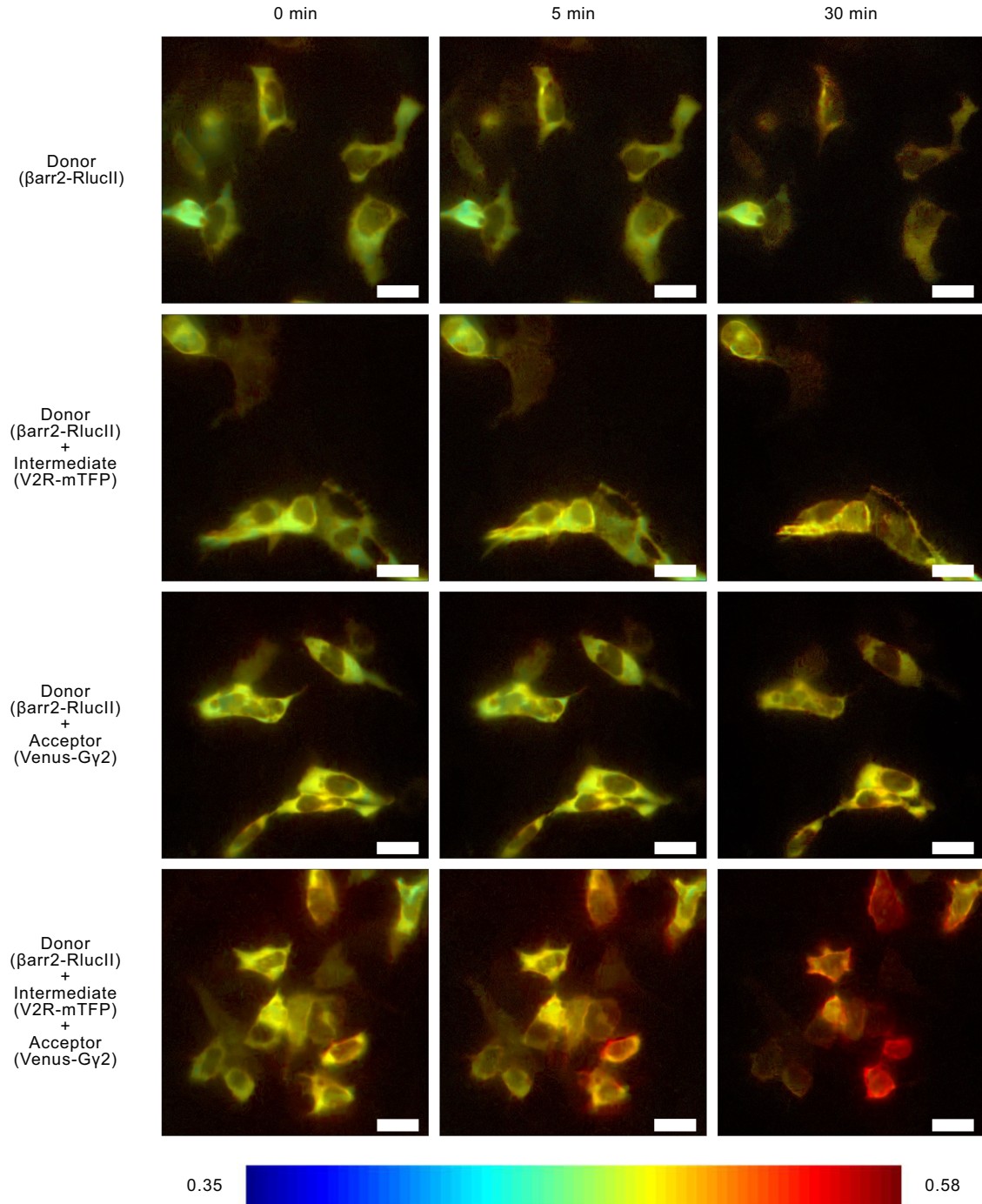

**Fig. 4 | V₂R–βarr2–Gβγ complex formation monitored by BRETfect microscopy.** Co-expression of BRETfect constructs βarr2-RlucII, V₂R-mTFP and Gγ2-YFP in parental HEK293T followed by 100 nM AVP stimulation for 5 and 30 min and image acquisition by luminescence microscopy. The numeric scale of the heat-map legend represents calculated BRET ratios. Scale bar: 5 μm.

## Molecular determinants of the Gβγ–βarr interaction

Structural comparison of Gβγ bound to three effectors: GRK2, GIRK2, and phosducin, revealed that Gβγ typically binds its effectors via its inner toroidal surface (Fig. 7a)[54–56]. This same surface is occupied by the GDP-bound Gαs within the heterotrimeric G$_s$[57], suggesting that G protein activation by receptor is critical in freeing up Gβγ and allowing for Gβγ–βarr association. With this in mind, we asked whether Gβγ binds preferentially to either inactive or active forms of βarr. To biochemically test the ability of Gβγ to directly associate with βarr1, we performed in vitro pull-down between purified Gβγ and GST-tagged βarr1, which shows that Gβγ binds to βarr1 in its inactive conformation (Fig. 7b, c). Subsequently, to test if Gβγ can also bind to βarr1 in its active conformation, we additionally performed a pull-down between purified Gβγ and the Flag-tagged β₂V₂R–βarr–Fab30 complex in the presence of the β₂AR agonist BI-167107. Gβγ was found to also associate with βarr in the context of a GPCR–βarr complex, indicating that the activated βarr associated to the receptor can also bind Gβγ (Fig. 7d, e).

To quantitatively probe the Gβγ–βarr interaction, we employed isothermal titration calorimetry (ITC) to obtain binding constants between Gβγ and various forms of active or inactive βarr. Gβγ binds to inactive βarr1 with an affinity of 6.1 μM (Fig. 8a). Similarly, Gβγ binds specifically to an

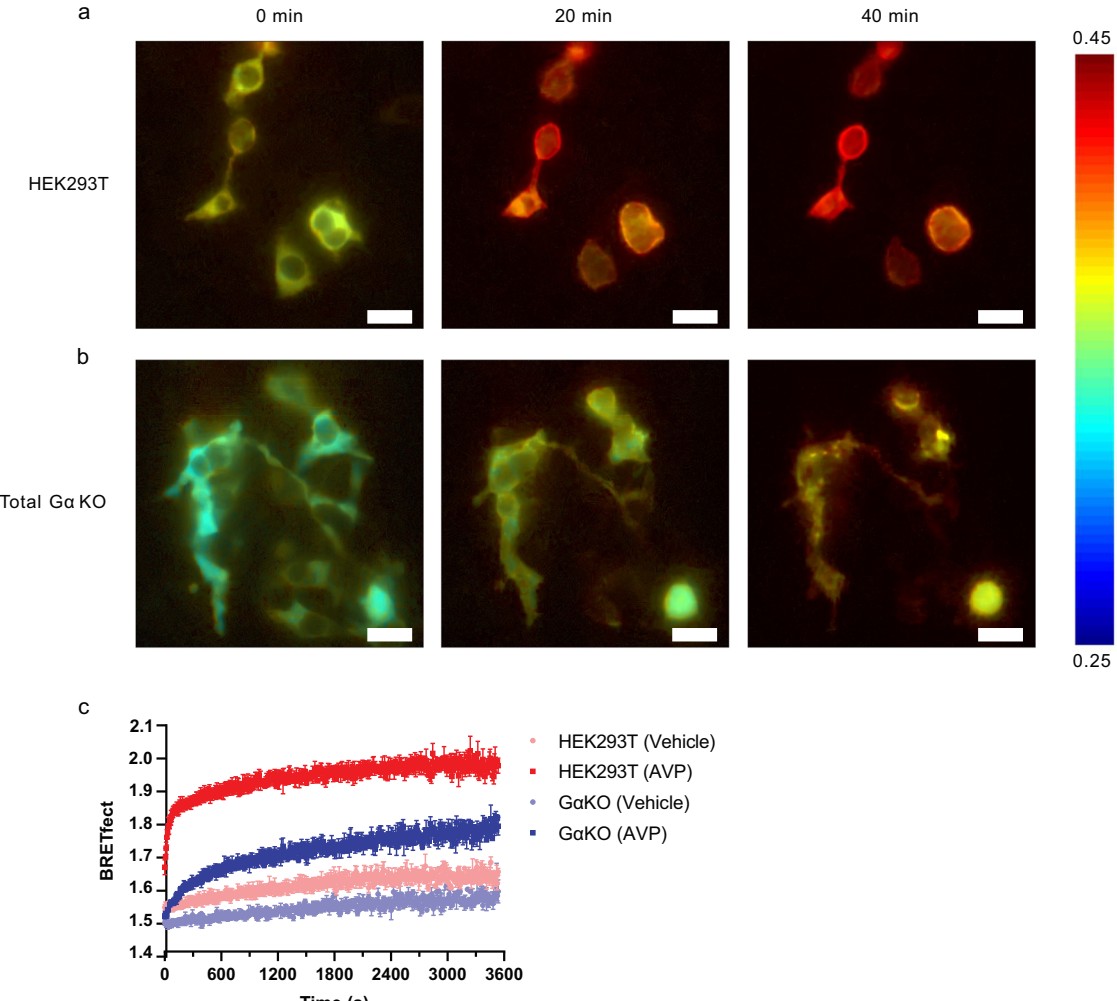

**Fig. 5 | V₂R–βarr2–Gβγ complex formation monitored by BRETfect microscopy and kinetics in parental HEK293T and Gα KO cells.** Co-expression of βarr2-RlucII, V₂R-mTFP and Gγ₂-YFP in parental HEK293T cells (**a**) or in total Gα KO cells (**b**) followed by 100 nM AVP stimulation and image acquisition by luminescence microscopy. The numeric scale of the heat-map legend represents calculated BRET ratios. **c** Co-expression of βarr2-RlucII, V₂R-mTFP and Gγ₂-YFP in parental HEK293T cells or total Gα KO cells followed by vehicle or 100 nM AVP treatment and BRETfect reading. Data are represented as the mean ± SEM ($n = 3$) (data in supplementary data files). Scale bar: 5 µm.

active βarr1–V₂Rpp–Fab30 complex, where V₂Rpp is a previously validated phosphorylated carboxy-terminal peptide derived from the human V₂R[58], with an affinity of 2.7 µM (Fig. 8b). Gβγ displayed the same propensity to bind to both inactive βarr2 (Fig. 8c) and βarr2 in the presence of 4-fold molar excess of V₂Rpp (Fig. 8d), with an affinity of 1.5 µM and 1.8 µM, respectively. An excess V2Rpp was used to estimate the affinity between Gβγ and βarr making sure that all βarr would be engaged by V2Rpp so to prevent having two populations of βarr that could influence the measured Kd. Using purified soluble Gβγ with a C68S point mutation in Gγ₂ that abrogates its prenylation site[59–61], we additionally show that soluble Gβγ maintains its capacity to bind to inactive βarr1 and βarr2 with affinities of 0.8 and 0.9 µM, respectively (Supplementary Fig. 7a, b). These experiments reveal that Gβγ is capable of specifically binding both βarr1 and βarr2 either in the inactive or active conformation, at low micromolar affinity without βarr conformational or subtype preference.

## Discussion

It has been long observed that some GPCRs continue to signal from within endosomes[21,62,63]. Second messenger molecules generated from subcellular compartments modulate enzymes within their immediate vicinity, potentially leading to differential physiological responses compared to those generated at the plasma membrane[64–67]. Recent works illustrate that the

related GPCR–βarr–Gₛ megaplex and GPCR–βarr–Gβγ complex provide a biophysical basis for said internalized signaling. Previous cellular experiments suggested an interaction between βarr and Gβγ, and that increases in free cellular Gβγ leads to enhanced cAMP generation from internalized compartments[21,26,29,68]. However, the mechanism by which βarr and Gβγ enhance endosomal G protein signaling remains unknown.

In this study, we demonstrate the formation of a ternary V₂R–βarr–Gβγ complex at the plasma membrane. The use of the BRETfect approach with various Gα KO cell lines allowed us to distinguish the V₂R–βarr–Gβγ complex from the previously shown V₂R–βarr–Gₛ megaplex (Fig. 3). Our findings suggest that βarr is required for Gβγ trafficking to the endosomes (Fig. 1e-f). Notably, and consistent with previous observations[34,36–39], receptor-activated Gαₛ dissociates from the plasma membrane, translocates to the cytoplasm independently of βarr and translocates to endosomes following Gβγ to this subcellular compartment. Indeed, Gβγ scavenging via plasma membrane-tethered βARKct not only greatly attenuated V₂R–βarr–Gβγ complex formation but also largely decreased Gαₛ endosomal trafficking. Interestingly, endosome-tethered βARKct did not impair V₂R–βarr–Gβγ complex formation, likely because the complexes have preformed at the plasma membrane and thus are more resistant to Gβγ sequestration once in the endosomes. Strikingly, our data indicates that the Gα and Gβγ subunits use different trafficking routes to

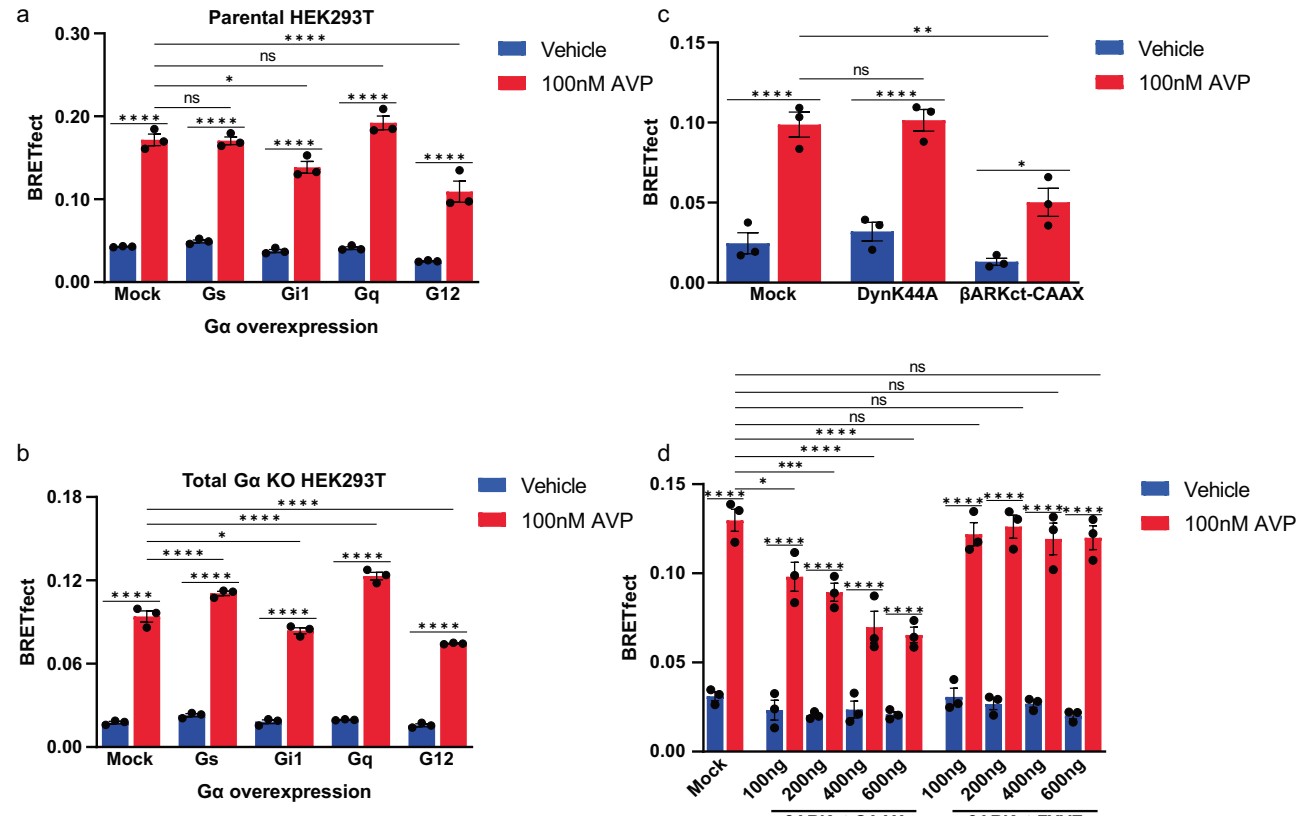

**Fig. 6 | GPCR and G proteins activation induces V₂R–βarr2–Gβγ complex formation at the plasma membrane.** AVP-promoted (100 nM) V$_2$R–βarr2–Gβγ complex formation monitored by BRETfect measurement between βarr2-RlucII, V$_2$R-mTFP and Gγ$_2$-YFP and co-expression of different Gα proteins (G$_s$, G$_{i1}$, G$_q$ and G$_{12}$) in parental HEK293T (**a**) or in Gα proteins depleted cells (**b**). **c** AVP-promoted (100 nM) V$_2$R–βarr2–Gβγ complex formation monitored by BRETfect measurement between βarr2-RlucII, V$_2$R-mTFP and Gγ$_2$-YFP and co-expression of internalization inhibitor DynK44A or plasma membrane anchored βARKct-CAAX

peptide. **d** AVP-promoted (100 nM) V$_2$R–βarr2–Gβγ complex formation monitored by BRETfect measurement between βarr2-RlucII, V$_2$R-mTFP and Gγ$_2$-YFP and co-expression of increasing amounts of plasma membrane anchored βARKct peptide (βARKct-CAAX) or endosomes anchored βARKct peptide (βARKct-FYVE). Data are represented as the mean ± SEM ($n = 3$) and statistical significance of the differences was assessed using a two-way ANOVA followed by Holm-Šídák's multiple comparison test (ns nonsignificant; *$P \leq 0.05$; **$P \leq 0.01$; ***$P \leq 0.001$; ****$P \leq 0.0001$).

reach the endosomes (Fig. 1). It should be noted that, because Gα$_s$ translocation was monitored by the BRET level between Gα$_s$-RlucII and rGFP-CAAX, we cannot exclude that part of the decrease BRET signal could originate from a redistribution of Gα$_s$ in the plasma membrane and not only from a dissociation from the plasma membrane.

Our data also suggest that the generation of free Gβγ through receptor activation of heterotrimeric G protein is a key event in V$_2$R–βarr–Gβγ complex formation, as overexpression of Gα proteins that do not couple productively to the V$_2$R significantly blunted complex formation, most likely by scavenging Gβγ and not allowing its dissociation upon V2R stimulation (Fig. 6a-b). Furthermore, total Gα knockout led to slower V$_2$R–βarr–Gβγ and reduced complex formation than in the presence of Gα subunits (Fig. 3, Fig. 5c, Fig. 6), further confirming the role of G protein activation in facilitating complex formation and indicating that a functional heterotrimer favors delivering Gβγ to the V$_2$R–βarr complex. This is corroborated by our biochemical characterization of promiscuous binding between free Gβγ and βarr1 and βarr2 without conformational or subtype preference (Fig. 8). A recent study suggests that βarr could be pre-associated at the plasma membrane[69], offering the possibility of constitutive binding of βarr with Gβγ even in the absence of agonist stimulation. This possibility that at least part of the βarr-Gβγ complex could be preformed and that the BRET increase between βarr-Rluc and Gγ-YFP could result from a conformational change. This could be comforted by the fact that both active and inactive βarr can spontaneously interact with Gβγ in vitro. Yet

the large translocation of βarrestin to the plasma membrane observed upon agonist stimulation (Fig. 5) indicates that an important part of the complex is agonist-promoted. We also cannot exclude the possibility that different population of the V2R could be bound to either βarr or Gβγ and brought in the same complex by receptor dimerization since V2R-dimer has been shown to bind to βarr and Gβγ[48]. Yet, such complex would be believed to internalize as a whole, and the overall mechanism of βarr-dependent G protein endocytosis described here would still hold.

Given that G protein and βarr were classically thought to function independently, our data adds to the growing body of evidence that these transducers operate co-dependently. Previous reports have demonstrated co-activation of multiple GPCRs or G protein pathways that synergistically potentiate sustained signaling via Gβγ-mediated mechanisms. Co-stimulation of both the G$_s$-coupled β$_2$-adrenergic receptor (β$_2$AR) and the PTHR led to synergistic increases in endosomal cAMP generation, mediated by Gβγ through direct modulation of adenylyl cyclase and prolonged interaction with βarr. In addition, activation of the G$_q$ pathway via PTHR stimulation enhanced cAMP generation through additional free Gβγ generation[29,68]. In line with a recent report demonstrating distinct subcellular localization of specific combinations of G protein β and γ subunits, we speculate that Gβγ subtypes may lead to Gα translocation to different intracellular compartments to facilitate distinct subcellular sustained signaling[33].

Taken together, our results point to a scenario whereby agonist stimulation of a GPCR leads to activation and dissociation between Gα and

**Fig. 7 | Gβγ binds to inactive and active βarr1 in vitro. a** Structural analysis of Gβγ-effector complex structures illustrate variable interaction between residues at the Gβγ inner toroidal surface with those of effectors. Each β sheet of Gβ is numbered 1 through 7. **b** In vitro pull-down between GST-βarr1 and Gβγ. **c** Quantification of GST-βarr1 and Gβγ pulldown. **d** In vitro pull-down between Flag-β₂V₂R–βarr1–Fab30 and Gβγ. **e** Quantification of Flag-β₂V₂R–βarr1–Fab30 and Gβγ pulldown. Data are represented as the mean ± SEM (*n* = 4) and statistical significance of the differences was assessed using an unpaired *t* test (**P ≤ 0.01; ***P ≤ 0.001).

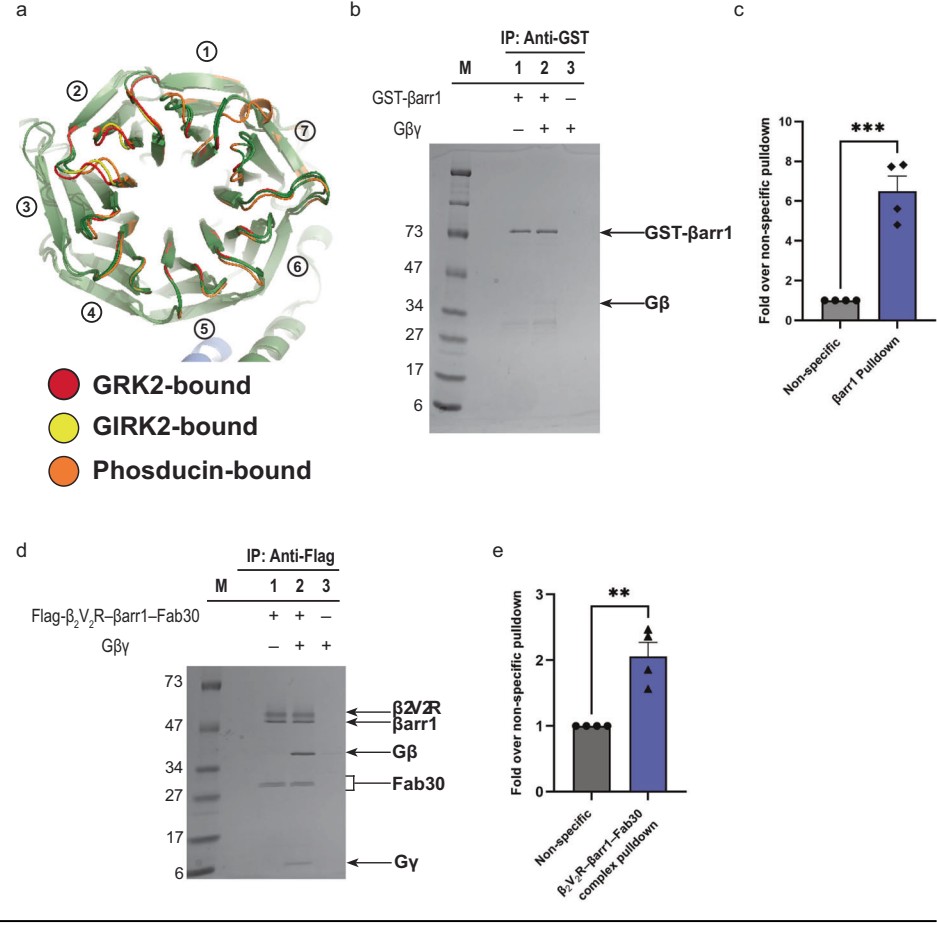

Gβγ, recruitment of βarr to a GRK-phosphorylated receptor, eventually forming a GPCR–βarr–Gβγ complex that is internalized into endosomes (Fig. 9). Activated and depalmitoylated Gα$_s$ dissociates from the plasma membrane forming a pool of cytoplasmic Gα subunits that probes endomembrane compartments. The presence of Gβγ in the endosomes spurs the translocation of Gα$_s$, thus regenerating competent G$_s$ heterotrimers and reassociation with an activated receptor in this compartment to hasten another round of second messenger molecule generation within subcellular compartments (Fig. 9). Our work therefore provides an explanation for how βarr and Gβγ contribute in a coordinated manner to sustained G protein signaling from within internalized compartments.

## Methods
### Reagents
Dulbecco's phosphate-buffered saline (PBS), Hanks' Balanced Salt Solution (HBSS), Dulbecco's modified Eagle's medium (DMEM), Trypsin, penicillin/streptomycin, fetal bovine serum (FBS), and newborn calf serum were purchased from Wisent Bioproducts. Polyethylenimine (PEI) was purchased from Alfa Aesar (Thermo Fisher Scientific). Arginine vasopressin (AVP) was from Sigma-Aldrich. Coelenterazine H and Prolume Purple were purchased from Nanolight Technologies. The V₂R phosphopeptide (V₂Rpp) was synthesized by the Tufts University peptide synthesis core facility.

### Cell lines
Parental HEK293SL and βarr1 and βarr2 KO cells were a gift from Dr Stephane Laporte (McGill University, Montreal, Quebec, Canada). HEK293T, Gα$_s$ KO and total Gα KO cells were a gift from Dr Asuka Inoue (Tohoku University, Sendai, Miyagi, Japan).

## Enhanced bystander bioluminescence resonance energy transfer
Cells were cultured in DMEM supplemented with 10% FBS, 100 units of penicillin, and 100 μg per ml streptomycin. Cells in suspension were transiently transfected at a density of 0.4 million cells per ml using 25 kDa linear PEI as a transfecting agent, at a ratio of 4:1 PEI:DNA.

For Gα$_s$ trafficking, parental HEK293SL and βarr1 and βarr2 KO cells were transfected with FLAG-V₂R, Gα$_s$67-RlucII (RlucII BRET donor fused at residue 67 of Gα$_s$[70,71] and rGFP-CAAX or rGFP-FYVE (BRET acceptor), and co-transfected with βARKct-CAAX, βARKct-FYVE, Gβ₁ and Gγ₂, as indicated in the figure's legends.

For Gβγ trafficking assays, parental HEK293SL and βarr1 and βarr2 KO cells were transfected with FLAG-V₂R or HA-CXCR4, Gγ₂-RlucII (BRET donor) and rGFP-CAAX or rGFP-FYVE (BRET acceptor). Cells are supplemented with βarr1 and βarr2 as indicated.

For receptor trafficking assays, parental HEK293SL and βarr1 and βarr2 KO cells were transfected with V₂R-RlucII or CXCR4-Rluc (BRET donor) and rGFP-CAAX or rGFP-FYVE (BRET acceptor). Cells are supplemented with βarr1 and βarr2 as indicated. DynK44A is co-transfected to block receptor internalization.

Transfected cells were seeded in 96-well microplates (Greiner) (100 μl per well). Forty-eight hours post-transfection, DMEM was removed, and cells were washed with PBS and replaced by HBSS. Cells were then treated with vehicle or agonists for the indicated time in the figure's legends and Prolume Purple (1 μM) was added for 6 min. BRET readings were done on a Spark multimode microplate reader (Tecan) equipped with a dedicated PMT—single photon counting Multi-color for BRET2 (400/70 nm (donor) and 515/20 nm (acceptor)). The BRET signal was calculated as the ratio of light emitted at the energy acceptor wavelengths over the light emitted at the

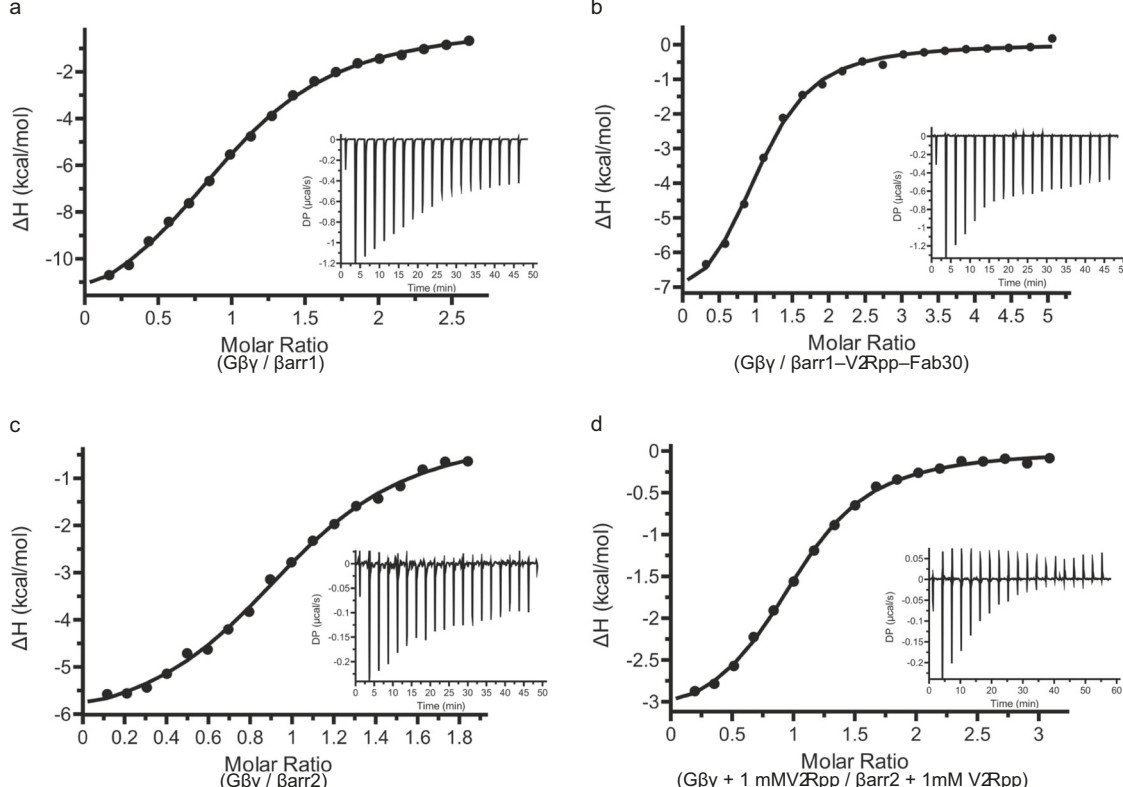

**Fig. 8 | Gβγ displays promiscuous, micromolar affinity binding against inactive and active βarr1 and βarr2.** Binding isotherm and thermogram (inset) between Gβγ and **a** inactive βarr1 ($K_D = 6.1 \pm 0.5$ μM; $N = 1.0$ site), **b** βarr1–V$_2$Rpp–Fab30 complex ($K_D = 2.7 \pm 0.4$ μM; $N = 1.0$ site), **c** inactive βarr2 ($K_D = 1.5 \pm 0.2$ μM; $N = 1.0$ site), and **d** βarr2 in excess V$_2$Rpp ($K_D = 1.8 \pm 0.2$ μM; $N = 1.0$ site). ITC data were fitted to a one-site binding model.

energy donor wavelengths. The agonist-induced BRET response is calculated by deducting the BRET signal obtained in the presence of vehicle from the BRET signal obtained in the presence of agonist.

### BRET with fluorescence enhancement by combined transfer (BRETfect)
For BRETfect assays, HEK293T, Gα$_s$ KO, or total Gα KO cells were transfected with βarr2-RlucII, V$_2$R-mTFP and Gγ2-YFP. Gα proteins, DynK44A and βARKct peptides are co-expressed in the indicated experiments. Transfected cells were seeded in 96-well microplates (Greiner) (100 μl per well). Forty-eight hours post-transfection, DMEM was removed, and cells were washed with PBS and replaced by HBSS. Cells were then treated with vehicle or 100 nM AVP for 20 min and Coelenterazine H (2.5 μM) was added 10 min before reading on a Mithras LB940 photon-counting plate reader (Berthold Technologies) equipped with donor filter (480/20 nm) and acceptor filter (530/20 nm). The BRETfect signal was calculated as the ratio of light detected at the acceptor wavelengths over the light emitted at the energy donor wavelengths from which the signal calculated from the donor-only condition was subtracted.

### BRETfect microscopy
Microscopic imaging of BRET signals was performed with an inverted microscope (Eclipse Ti-E, Nikon), and EMCCD camera (HNu512, Nuvu Cameras)[50]. HEK293T cells were seeded on 35 mm glass bottom dishes and transfected with the BRETfect constructs for 48 h. Cells were washed with HBSS. Luciferase substrate (Coelenterazine H, 10 μM) was diluted with HBSS and added just before the measurement. Binary photon counting frames were continuously recorded with 100 ms exposure. Filter before the camera was switched every 10 s (100 frames) to alternately obtain total luminescence frames (without filter) and acceptor frames (with 510 nm long-pass filter). Final images were obtained by integrating the

same numbers of total luminescence frames and acceptor frames until the average photon count of the total luminescence image reaches 100 counts per pixel. BRET image was obtained by dividing acceptor photon counts by total photon counts, pixel by pixel. To reduce the shot noise level, BM3D filter adapted for Poisson noise reduction[72] was applied and contrast was slightly compressed (gamma = 1.5) for all BRET images. BRET level was described using pseudocolor allocated with "jet" color-map of MATLAB 2021b[73].

### Protein purification
For in vitro pulldown experiments, Gβγ was purified from bovine brain and the Flag-tagged β$_2$V$_2$R–βarr–Fab30 complex was purified using a Flag M1 affinity resin[9,74].

For isothermal titration calorimetry experiments, recombinant purified WT Gβ$_1$γ$_2$ was used[57,61]. Gβ$_1$γ$_2$ with a C68S mutation in Gγ$_2$ was generated using the Quikchange method (Agilent)[61]. Finally, purification of the following proteins was done as follows: GST-βarr using GST-pull down[75], the untagged βarr1 and βarr2 was generated by cleaving GST with thrombin, the βarr1–V$_2$Rpp–Fab30 complex using size-exclusion chromatography[58] and the Flag-tagged, BI-167107-occupied β$_2$V$_2$R–βarr1–Fab30 complex using Flag M1 affinity resin[9]. V$_2$Rpp was previously reported and synthesized by the Tufts University Peptide Synthesis Core Facility[76].

### Structural comparison of Gβγ–effector complexes
Previously published structures of Gβγ bound to various effectors, in this case, G protein-gated inward rectifier potassium channel 2 (GIRK2; PDB: 4KFM), GPCR kinase 2 (GRK2; PDB: 1OMW), and phosducin (PDB: 2TRC) were visualized in PyMol and aligned by their G protein beta subunits[54–56]. Subsequently, the interface between Gβ and the effectors was calculated using the InterfaceResidues script within PyMol.

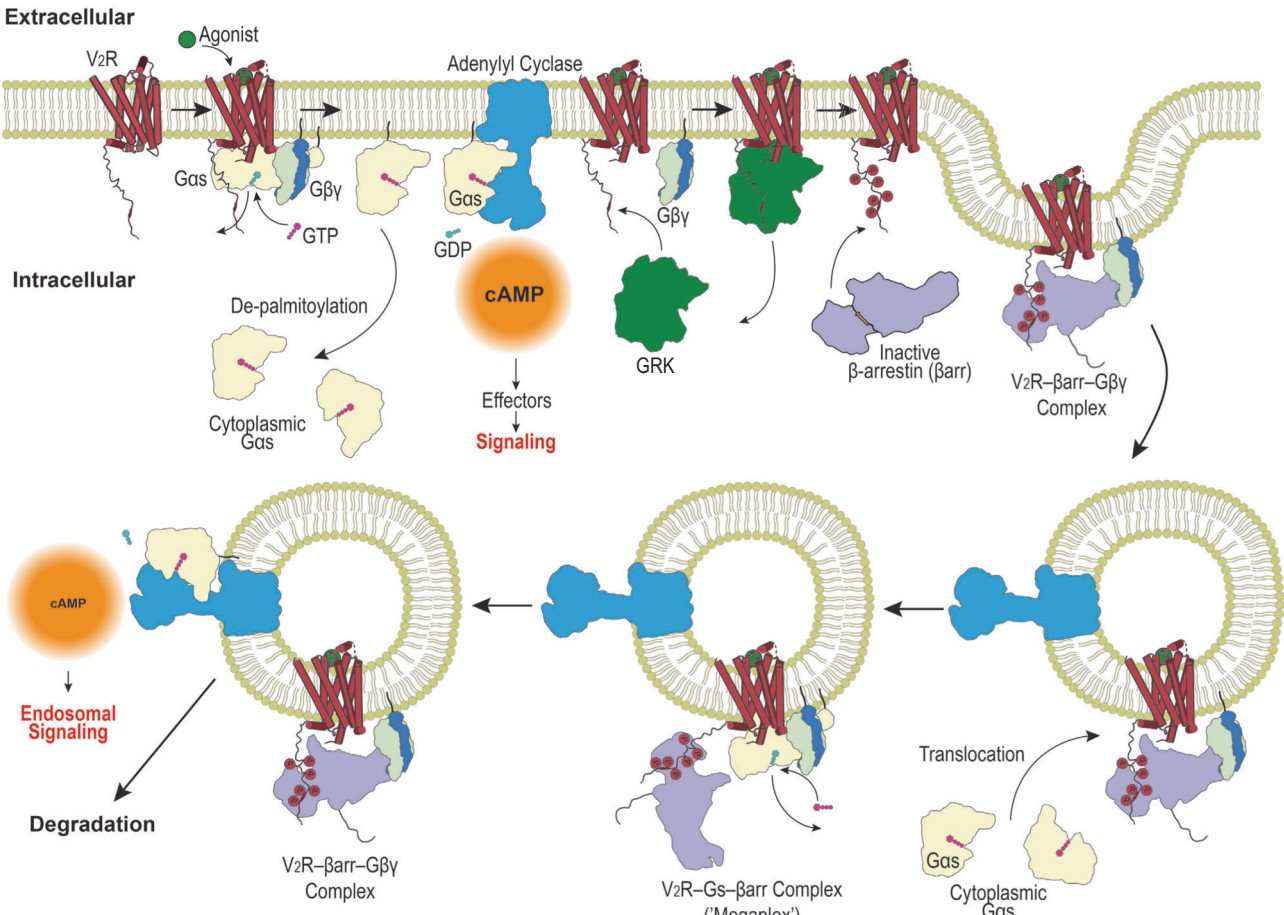

**Fig. 9 | Schematic illustrating mechanism of sustained endosomal G protein signaling.** Upon receptor activation, Gαs is depalmitoylated and dissociates from the plasma membrane. Concurrently, a V2R–βarr–Gβγ complex forms at the plasma membrane and undergoes endocytosis, allowing the reformation of endosomal heterotrimeric G protein and the generation of cAMP within the endosomes.

### In vitro pull-down

Flag-tagged, BI-occupied β2V2R–βarr1–Fab30 complex was mixed with Gβγ from bovine brain in a 1:3 ratio in an assay buffer containing 20 mM HEPES, pH 7.4, 150 mM NaCl, 0.01% LMNG, 100 nM BI and left to incubate for 30 min. Next, M1 anti-FLAG agarose beads and 2 mM $CaCl_2$ was added followed by another 30 min incubation. Subsequently, the beads were washed five times using the same assay buffer + 2 mM $CaCl_2$. The protein was eluted using an elution buffer containing 1 mg per ml FLAG peptide (Sigma-Aldrich), 20 mM HEPES, pH 7.4, 150 mM NaCl, 0.01% LMNG, 100 nM BI, 5 mM EDTA. Eluted samples were visualized by gel electrophoresis and quantified by ImageJ.

Similarly, GST-βarr1 was mixed with Gβγ from the bovine brain in a 1:3 ratio in an assay buffer containing 20 mM HEPES, pH 8.0, 100 mM NaCl, 0.01% DDM and left to incubate for 30 min. Next, glutathione Sepharose beads (GE Healthcare) was added followed by another 30 min incubation. Subsequently, the beads were washed five times using the same assay buffer and eluted with an elution buffer comprising 20 mM HEPES, pH 8.0, 100 mM NaCl, 0.01% DDM, 5 mg per ml reduced glutathione, 5 mM DTT. Eluted samples were visualized by gel electrophoresis and quantified by ImageJ.

### Isothermal titration calorimetry (ITC)

ITC measurements were performed using the MicroCal PEAQ-ITC (Malvern Panalytical). Purified βarr1, βarr2, or Gβγ were dialyzed overnight in 20 mM HEPES, 150 mM NaCl, 0.02% DDM, pH 7.4. The dialysis buffer was subsequently used to wash each component of the ITC instrument. ITC experiments were performed with six distinct conditions: (1) 30 μM of βarr1 in the sample cell and 400 μM of Gβγ in the injection syringe, (2) 15 μM βarr1–V2Rpp–Fab30 in the sample cell and 400 μM Gβγ in the injection syringe, (3) 12.5 μM of βarr2 in the sample cell and 125 μM of Gβγ in the injection syringe, (4) 15 μM of βarr2 with 1 mM V2Rpp in the sample cell and 250 μM of Gβγ with 1 mM V2Rpp in the injection syringe, (5) 15 μM of βarr1 in the sample cell and 400 μM of Gβγ with a C68S mutation on Gγ2 (Gβγ C68S) in the injection syringe, and (6) 20 μM of βarr2 in the sample cell and 400 μM of Gβγ C68S in the injection syringe. The sample cell was equilibrated to 25 °C, the reference power was set to 5.5 μcal s$^{-1}$ and the sample cell was stirred continuously at 750 rpm. Each titration experiment was initiated by a 0.4 μL injection from the syringe, followed by eighteen 2.0 μL injections at 180 s intervals. Raw data excluding the first injection were baseline corrected, and each peak area was integrated and normalized. Data was analyzed using the MicroCal PEAQ-ITC analysis software (version 1.41) and fitted to a one-site nonlinear least-squares fit model to obtain a dissociation constant ($K_D$). Representative isotherm and binding thermogram, of two independent experiments, are shown. Both experiments involving Gβγ C68S against βarr1 and βarr2 appear to have an additional early binding site, which was excluded, and the rest of the data were fitted to a one-site model.

### Statistical analysis

All the data in the manuscript are presented as means ± S.E.M from experimental replicates indicated in the figure legends. The type of statistical analysis and post-hoc test used are included in each figure legend. We have used GraphPad PRISM version 9 and considered a p value of <0.05 as statistically significant.

## Reporting summary

Further information on research design is available in the Nature Portfolio Reporting Summary linked to this article.

## Data and materials availability

All data needed to evaluate the conclusions in the paper are present in the paper and/or the Supplementary Materials. All the source data can be found in the "Supplementary Data" file associated with the manuscript. Some of the biosensors used in the present study are protected by patents, but all are available for academic research under regular material transfer agreement upon request to M.B.

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

## Acknowledgements

We thank Robert Lefkowitz for helpful discussion and thoughtful feedback, as well as for experimentation support through NIH grant R01HL016037. This work was supported by NIH grant F30HL149213 (A.H.N.) and a Foundation Grant from the Canadian Institute for Health Research FDN-148431 as well as a Natural Sciences and Engineering Research Council of Canada grant RGPIN/05556-2019 (MB). B.S. was financially supported by a scholarship from Fonds de Recherche du Québec – Santé (FRQS). A.H.N. was additionally supported by a fellowship through the HHMI Medical Research Fellows Program.

## Author contributions

B.S., A.H.N., A.R.B.T., and M.B. conceived the project and designed experiments. B.S. and H.K. performed BRET and BRETfect experiments.

A.H.N, A.R.B.T, L.-Y.H., A.W.K., J.K., B.H., S.M., J.L.IV, C.E., I.P., and E.H. purified protein, performed structural analysis, in vitro pull-down, and ITC experiments. B.S., A.H.N, A.R.B.T., and M.B. wrote the manuscript. M.B. supervised the project.

## Competing interests

M.B. is the president of the scientific advisory board of Domain Therapeutics, a biotech company to which some of the biosensors used in this study were licensed for commercial use. All other authors declare no competing interests.
