## [Peer Review File · Communications Biology]

Reviewers' comments:

Reviewer #1 (Remarks to the Author):

The manuscript by Sokrat et al reports strong evidence for the existence of a ternary complex between the vasopressin V2 receptor (V2R), beta-arrestin (barr) and the G β g subunit (G β g) upon agonist stimulation (the arginine-vasopressin (AVP) ligand). In the field of G protein-coupled receptor (GPCR) molecular pharmacology, V2R is generally used as a model GPCR that maintains a sustained interaction with barr during internalization. It allows these receptors to continue signaling within internalized compartments such as endosomes. V2R is well known to couple and activate the trimeric G protein Gs. In this study, the authors attempt to demonstrate that the V2R-barr-G β g ternary complex is involved in the translocation of Galpha-s and G β g subunits to endosomes. Interestingly, trafficking of Galpha-s and G β g subunits to endosomes is lower in a barr KO cell line. The authors use biochemical and biophysical approaches such BRET technologies, including the BRETfect method that has recently been reported to monitor ternary protein complexes (Cotnoir-White et al (2017) PNAS).

Although the study presents interesting data, several points need to be clarified.

The main concerns are :

- The title gives a strong conclusion that is not fully supported by the results. The message of the title should appear more as one possibility among others than as a definitive conclusion.
- How can we be sure that the ternary complex is not constitutively formed in the transfected cells, and that the agonist has merely induced a conformational change in this complex? Figure 8 shows that the direct interaction between barr and G β g does not depend on the conformation (active or inactive) of barr.
- Figure 1 : free G β g increases translocation of Galpha-s from the plasma membrane (PM) to endosomes. But why does overexpression of free G β g have a greater effect on increasing Galpha-s dissociation from the PM (Fig. 1A) than on increasing Galpha-s in endosomes (Fig. 1C) ?
- Figure 4 : since the formation of the V2R-barr-G β g ternary complex occurs rapidly after stimulation, why are the images not provided at a shorter time (before 5 min) ?
- Figure 5 : why not show results with Galpha-s KO since V2R is mainly coupled to Gs proteins ?
- Figure 5B : V2R-barr-G β g ternary complex formation appears very low in Galpha total KO cells (and does not appear to occur at the PM), which seems in contrast to the BRETfect results measured in Figure 3D where Galpha total KO conditions does not appear to impair ternary complex formation as strongly.
- The authors do not discuss the possibility of having different populations of receptors bound to either barr or G β g, at the same time point. The different receptor populations could be close and even form dimers or oligomers, as the authors recently proposed in the study reporting the development of the BRETfect method (Cotnoir-White et al (2017) PNAS).
- Related to this specific point, in Figure 9 what about the possibility of having barr or G β g bound to a V2R dimer or oligomer as the authors have recently proposed (Cotnoir-White et al (2017) PNAS)?
- Figure 6A-B: these data are the most critical. How to discriminate between the two possible effects of overexpressing Galpha-i and -12 subunits : 1) reducing the number of V2R-barr-G β g

ternary complexes ; 2) their influence on the conformation of the ternary complex, without destabilizing it.

- Figure 6 : which Gi proteins are used ?

Minor points:

- Legend figure 1E-F: "beta1/2" is unclear. Does this mean that beta-arrestin-1 and beta-arrestin-2 have been cotransfected? The same applies to the title of Figure 8.

- Legend figure 3D: could you indicate which Galpha subunits are missing?

- Fig. 8C: why is an excess of V2Rpp necessary?

Reviewer #2 (Remarks to the Author):

The manuscript by Sokrat et al describes the formation and role of β arr and $G\beta\gamma$ in sustained GPCR signalling. The manuscript is well written with clear description of the methods and results as well as relevant discussion. The manuscript uses a range of techniques to demonstrate the formation, trafficking and regulation of β arr- $G\beta\gamma$ complexes. While the functional consequence of formation of these complexes is not investigated in the present manuscript. The discussion on this matter is logical and with only minimal speculation. Moreover, this manuscript provides key mechanistic insight to our understanding of sustained GPCR signalling from endosomes.

I have the following comments on the content of the manuscript for the authors to consider.

Major comments:

I have major concerns regarding the statistical analyses used throughout the manuscript both in terms of detail and tests used. I was unable to find in the methods section details on the statistical tests used nor any justification for their use.

a. The figure legends do contain the information that the p-values stated are "(unpaired t-test)" although this appears to be a boiler plate sentence and is sometimes included even where no statistical analysis is indicated on the graph e.g. SuppFig3. It would be useful for interpretation of the data for the tests used, with justification, to be outlined in the manuscript.

b. The authors should justify their use of an unpaired t-test without correction for multiple comparisons throughout the manuscript. Indeed, for many analyses/graphs an ANOVA appears to be more appropriate.

c. From viewing the graphs, it is sometimes unclear why some comparisons were made and another not. For example, Fig2a it appears statistical analysis was performed on the Mock Barr1/2 KO group but not the mock parental group, yet a reduction was also seen in the parental group? In Figure 3B comparisons are made in the text between the vehicle and AVP-induced responses "In parental HEK293T cells (Fig. 3B), expression of β arr2-RlucII with V2R-mTFP (D + I) resulted in an AVP-induced increase in signal, indicative of recruitment of β arr2 to the receptor. Expression of β arr2-RlucII with Gy2-YFP (D + A) did not result in an agonist-induced response in the absence of

overexpressed V2R,...” yet there are no comparisons made on the graph.

d. Some graphs indicate significance with asterisk that are not linked to another group making assessment of the comparison impossible without further detail in the figure or graph. Again, notable examples are Fig2a but also Fig 2b, Figure 6d, SuppFig 2. Which groups are being compared should be made clear either on the graph or figure legend.

Minor comments

1 pg6: The authors state that “release from the plasma membrane was found to be restricted to this compartment since anchoring β ARKct to the endosomes using the FYVE targeting domain of endofin” Yet in Figure 1b FYVE targeted β ARKct appears to significantly reduce the change in BRET in the presence of additional $\beta\gamma$. Can the authors comment on a mechanism of action for this observation.

2. pg10, The authors present data on the role of G α subunits in the formation of V2R– β arr–G $\beta\gamma$ complexes (Fig6). They also note that such complexes can exist “as a unique entity without the incorporation of G α subunits.” based on the data in Fig 3. While not contradictory there does appear to be a decrease in the BRETfect ratio between the Fig3B and Fig3D for the AVP induced response potentially further supporting the authors observations in Fig 6. The authors may wish to comment on this.

3. Supplementary Figure 1: It is not clear from the methods or figure legend the difference between D+I and D+I+A in S1 and Fig3B? As interpreted the figure legend indicates that only D+A has the addition of V2R. Are these the same data or additional replicates? If additional replicates without the addition of unlabelled V2R it is unclear what the addition of the D+I and D+I+A groups adds to the manuscript. There are also no p-values indicated on the figure.

Reviewer #3 (Remarks to the Author):

The manuscript by Badr Sokrat#, Anthony H. Nguyen#, et al. entitled “The V2R– β arrestin–G $\beta\gamma$ complex promotes G protein translocation to endosomes” Describes the dissociation of the G α subunit from the active G-protein at the plasma membrane and obvious re-association with the receptor on endosomes. This is a nice and timely study and well written with easy to follow arguments backed by several well thought control experiments.

Introduction:

- Clearly written! Therefore, I only have very minor points
- Line 57/58: I would suggest to use “expected of a receptor-activated β arr” instead of “expected of an activated β arr” to be more specific since there may be other active states of arrestin e.g. binding to JNK3 that favor a different state of arrestin but this arrestin is still active....
- Line 64-67: “Several class B GPCRs such as the parathyroid hormone receptor (PTHr), neurokinin 1 receptor (NK1R) and the vasopressin type 2 receptor (V2R) have been shown to continue signaling

within internalized compartments instead of remaining desensitized.” Please indicate the appropriate references for these receptors and finding.

Results section:

- “Gas dissociates from the plasma membrane after V2R activation and translocates to endosomes”:

o Could the authors please reason why they used the time-point of 20 min to look at AVP-induced Gas dissociation from the plasma membrane? Is this a value taken from literature? Did the authors check the signal over time and then decided on this time point as it reaches a stable plateau? In that case it would be great to see the measurement over time in the supplements. This would be nice/helpful to know for the readers.

o Please show that the $G\alpha$ and $G\beta\gamma$ actually dissociate to make clear that it is really reconstitution of the heterotrimers that the authors measure in this assay! (Also would add more confidence to lines 139-141 “Endosomal $G\beta\gamma$ could then attract the Gas released from the plasma membrane, allowing the reconstitution of a trimeric G protein in the endosomal compartment.”)

o Does the overexpression of the full GRK2 lead to the same effect? This would be an interesting add on to contextualize the finding and show how the full GRK2 protein influences this mechanism as it has been shown that GRK2 expression varies in different cell types but also in pathophysiological conditions!

o Lines 112-114: “These results suggest that the formation of heterotrimeric Gs and dissociation of Gas from the plasma membrane is dependent on the presence of free $G\beta\gamma$.” As the formation of heterotrimeric Gs is not directly measured here, shouldn’t the phrasing of the causality rather be “These results suggest that the dissociation of Gas from the plasma membrane is dependent on the presence of free $G\beta\gamma$ and the formation of heterotrimeric Gs.”? Because the presence of the inhibitory $G\beta\gamma$ scavenger β ARKct-CAAX reduces and the increased availability of free $G\beta\gamma$ by overexpression drastically enhances the dissociation of Gas from the plasma membrane, hence the interaction with $G\beta\gamma$ and the formation of a heterotrimer is probably necessary for this process.

o Lines 114- 118: “The effect of scavenging free $G\beta\gamma$ with β ARKct on Gas release from the plasma membrane was found to be restricted to this compartment since anchoring β ARKct to the endosomes using the FYVE targeting domain of endofin44 had little impact on Gas dissociation from the plasma membrane with or without $G\beta\gamma$ overexpression (Fig. 1B).” The dissociation under conditions when $G\beta\gamma$ is overexpressed is significantly reduced. How would the authors interpret that?

o In line lines 120, 121: the authors write “suggesting that β arr is not required for Gas dissociation from the plasma membrane (Fig. 1A).”; lines 129-131 “Strikingly, we observed a significant reduction in Gas translocation to endosomes in β arr1/2 KO cells (Fig. 1C), whereas Gas dissociation from the plasma membrane was not impaired by β arr1/2 depletion (Fig. 1A).”; lines 133, 134: “However, depletion of β arr only impairs Gas endosomal translocation, not its dissociation from the plasma membrane.”

This is quite confusing since the reduction of BRET signal in the CAAX assay might not actually reflect a dissociation from the plasma membrane but rather re-distribution processes within the membrane that are happening independently from β arr1/2 while the actual dissociation from the plasma membrane and trafficking to early endosomes is significantly dependent on β arr1/2. Please

consider re-phrasing this section a bit or comment on where the authors would suggest the Gas goes if it does not reach the early endosomal stage in β arr1/2 KO cells?

- “ β arr mediates $G\beta\gamma$ trafficking from the plasma membrane to the endosomes”:

o Please include a measurement of V2R-RlucII and CXCR4-Rluc to rGFP-FYVE to show that the receptors reach the endosomes and that this is not just an observed process in the membrane.

o “In contrast to what is observed for the V2R, for which the loss of plasma membrane receptor upon activation requires β arr (Fig. 2A).” There still seems to be some residual signal left (around 10%) in β arr-KO cells, even if the amplitude is very much reduced compared to the CXCR4. Could the author explain what this process is? 10% signal with such small error bars is not nothing....

- “V2R, β arr and $G\beta\gamma$ form a complex in cells”

o The authors state in lines 197, 198 “As seen in Fig. 3D, V2R- β arr2- $G\beta\gamma$ complex could still form in the total absence of $G\alpha$ proteins.” Could the authors please comment/explain the loss of measured BRETfect signal amplitude in total $G\alpha$ KO cells? Could it be that the presence of $G\alpha$ somehow enhances the effect/makes the process more efficient? \diamond lines 213-215 “Although the formation of the V2R- β arr2- $G\beta\gamma$ complex was also observed in the total $G\alpha$ KO, the formation kinetics was much slower ($t_{1/2}$: 288.8 sec) in the absence of $G\alpha$ subunits (Fig. 5B-C).” This fits quite well with the observation mentioned above for Fig. 3D?! Please comment.

- “Agonist-promoted V2R- β arr2- $G\beta\gamma$ complex formation occurs at the plasma membrane”

o Lines 225, 226: “In contrast, over-expression of $G\alpha_i$ and $G\alpha_{12}$, blunted complex formation (Fig. 6B).” “Blunted” seems like a strong word here. The signal reduction is visible and the signal is clearly diminished but blunted.

- Figure 1:

o Which AVP concentration was used to stimulate the receptor? Please definitely include this information in the figure legend and potentially also method section!

o In panel C the lower labeling seems to be cut off, please make sure that this is not the case in the final figure so that every label is legible.

o Might be helpful to include panel headings to make it easier to follow for the readers and make directly clear what they are looking at. E.g. “Gas dissociation from PM” for panel A

o Legend for A): “Overexpression of β ARKct-CAAX and $G\beta_1\gamma_2$ modulates Gas dissociation from the plasma membrane.” Instead of “Overexpression of β ARKct-CAAX and $G\beta_1\gamma_2$ modulates Gas dissociation.” to be more specific.

o Panel B, D: Please label “ $\beta\gamma$ ” as “ $G\beta\gamma$ ” as in the rest of the figure and manuscript to be consistent.

- Figure 4:

o I am aware that BRET microscopy is a great achievement but could the resolution of the images be improved? (Also for images in Fig.5)

o Could the authors provide some kind of quantification that is the underlying basis for the coloring? Maybe as a supplementary table?

- Figure 5:

o I don't want to be picky but to put the signal scale in the middle of the figure and have time traces follow with the same colors is more than misleading. I would suggest to put the scale bar to the right of the cell pictures.

- Figure 9:

o Why do the authors indicate GRK2 specifically in this overview? The authors did not test for this. What do the authors refer to? Do they want to indicate that there are isoform-specific differences? Otherwise the authors could just leave out the specification of the number 2.

Discussion:

- There is indication for bArr1 interaction with GNB1 in the literature, roughly localized to amino acids 180-280 in bArr1, as listed by Crepieux et al 2017 and found in Yang, 2009, *Biochem. J.*, 417, 287. This information could be added for the reader.

- Generally, it would be very helpful to follow the argumentation of the authors if they could include the appropriate reference to their results figure in the statements they make in the discussion about their results, so that it is easier for the readers to follow the line of thought.

Methods:

- I know this Gas67 has been around for quite some time but could the authors mention for the younger researchers what exactly the Gas67 construct is and what the 67 means, so that it is clear what kind of constructs were used in this study (Older work still deserves to be cited).

Reviewers comments:

Reviewer #1 (Remarks to the Author):

The manuscript by Sokrat et al reports strong evidence for the existence of a ternary complex between the vasopressin V2 receptor (V2R), beta-arrestin (β arr) and the Gbetagamma subunit ($G\beta\gamma$) upon agonist stimulation (the arginine-vasopressin (AVP) ligand). In the field of G protein-coupled receptor (GPCR) molecular pharmacology, V2R is generally used as a model GPCR that maintains a sustained interaction with β arr during internalization. It allows these receptors to continue signaling within internalized compartments such as endosomes. V2R is well known to couple and activate the trimeric G protein Gs. In this study, the authors attempt to demonstrate that the V2R- β arr- $G\beta\gamma$ ternary complex is involved in the translocation of $G\alpha$ -s and $G\beta\gamma$ subunits to endosomes. Interestingly, trafficking of $G\alpha$ -s and $G\beta\gamma$ subunits to endosomes is lower in a β arr KO cell line. The authors use biochemical and biophysical approaches such BRET technologies, including the BRETfect method that has recently been reported to monitor ternary protein complexes (Cotnoir-White et al (2017) PNAS).

Although the study presents interesting data, several points need to be clarified.

The main concerns are:

- The title gives a strong conclusion that is not fully supported by the results. The message of the title should appear more as one possibility among others than as a definitive conclusion.

Our title did not suggest that the V2R- β arrestin- $G\beta\gamma$ complex was the only mechanism allowing translocation of G proteins from the plasma membrane to the endosomes, but to make it clearer we modified it to indicate an involvement of this complex, that does not exclude other potential mechanisms. The new title is: Role of the V2R- β arrestin- $G\beta\gamma$ complex in promoting G protein translocation to endosomes.

- How can we be sure that the ternary complex is not constitutively formed in the transfected cells, and that the agonist has merely induced a conformational change in this complex? Figure 8

shows that the direct interaction between β arr and $G\beta\gamma$ does not depend on the conformation (active or inactive) of β arr.

It is indeed difficult to distinguish between an agonist-induced complex formation and an agonist-induced conformational change. Yet, most of the β arr translocation to the receptor in cells is mediated by the agonist-occupied receptor and it would be expected that β arr would largely not be present at the plasma membrane under basal conditions. This been said, a recent paper by David Calebiro's group (Grimes J. et al, Cell, 2023) suggests that a population of β arr could already be pre-associated at the plasma membrane. Moreover, the constitutive activity of the transfected receptor could promote β arr recruitment in the absence of agonist stimulation. It is therefore difficult to formally exclude that at least part of the complex could be constitutively formed, and that the agonist promotes a change in conformation. This has now been addressed in the discussion (line 323-330).

- Figure 1 : free $G\beta\gamma$ increases translocation of $G\alpha$ -s from the plasma membrane (PM) to endosomes. But why does overexpression of free $G\beta\gamma$ have a greater effect on increasing $G\alpha$ -s dissociation from the PM (Fig. 1A) than on increasing $G\alpha$ -s in endosomes (Fig. 1C) ?

We understand the point raised by the reviewer but unfortunately it is not possible to directly compare the absolute signals obtained with two distinct biosensors (rGFP-CAAX in one case and rGFP-FYVE in the other). Each biosensor has its own basal value and dynamic window.

- Figure 4 : since the formation of the V2R- β arr- $G\beta\gamma$ ternary complex occurs rapidly after stimulation, why are the images not provided at a shorter time (before 5 min) ?

We could detect V2R- β arrestin- $G\beta\gamma$ formation as early as 5 minutes post-stimulation but the signal was clearer at 30min due to signal accumulation. Yet, to address the reviewer's point, we added the images at 5 minutes to Figure 4 and the legend has been modified accordingly.

- Figure 5 : why not show results with $G\alpha$ -s KO since V2R is mainly coupled to Gs proteins ?

Since the V2R has been shown to also couple to Gq proteins (Avet C. et al., eLife, 2022), we used total G alpha protein KO to completely eliminate possible G alpha coupling. Yet, to

address the reviewer's suggestion, we added Fig. S5 where we show the formation of the V2R- β arrestin-G $\beta\gamma$ complex in Galpha-s KO cells. As was observed in the full G protein KO, the complex formation was slower than in WT cells. This new figure is now introduced on line 230 of the manuscript.

- Figure 5B : V2R- β arr-G $\beta\gamma$ ternary complex formation appears very low in Galpha total KO cells (and does not appear to occur at the PM), which seems in contrast to the BRETfect results measured in Figure 3D where Galpha total KO conditions does not appear to impair ternary complex formation as strongly.

The signal indicating V2R- β arrestin-G $\beta\gamma$ complex formation observed by microscopy in the Galpha KO cell is indeed lower compared to the parental cells. However, this is in line with the data in figure 3 as we can note lower BRETfect signal in the Galpha KO cells compared to the parental cells. The differences in the y axes may have led to the confusion. To avoid such confusion, we further emphasized this observation in the result section (line 214-215).

- The authors do not discuss the possibility of having different populations of receptors bound to either β arr or $G\beta\gamma$, at the same time point. The different receptor populations could be close and even form dimers or oligomers, as the authors recently proposed in the study reporting the development of the BRETfect method (Cotnoir-White et al (2017) PNAS).

- Related to this specific point, in Figure 9 what about the possibility of having β arr or $G\beta\gamma$ bound to a V2R dimer or oligomer as the authors have recently proposed (Cotnoir-White et al (2017) PNAS)?

The only way to directly address this possibility would be to inhibit V2R dimerization but no such tool currently exists. Thus, we have added in the discussion (line 330-334) that we cannot exclude the possibility that β arr and $G\beta\gamma$ complex could be formed through the formation of a V2R dimer.

- Figure 6A-B: these data are the most critical. How to discriminate between the two possible effects of overexpressing $G\alpha_i$ and $G\alpha_{12}$ subunits : 1) reducing the number of V2R- β arr- $G\beta\gamma$ ternary complexes ; 2) their influence on the conformation of the ternary complex, without destabilizing it.

At this point, we cannot differentiate between an overall reduction in complex formation and an effect of $G\alpha_i$ and $G\alpha_{12}$ on the conformation of the ternary complex. We can offer the hypothesis that $G\alpha$ subunits not productively interacting with V2R could inhibit the signal by titrating $G\beta\gamma$ away from the complex and prevent the engagement of β arr. We have added a sentence to suggest this possibility (lines 242-243). Yet, this does not detract from the main conclusion of the paper that V2R- β arrestin- $G\beta\gamma$ complex formation enable trafficking of G proteins from the plasma membrane to the endosomes.

- Figure 6 : which G_i proteins are used ?

The G_i subtype overexpressed is G_{i1} . We added this information to the figure panel, the legend and in the text.

Minor points:

- Legend figure 1E-F: "beta1/2" is unclear. Does this mean that beta-arrestin-1 and beta-arrestin-2 have been cotransfected? The same applies to the title of Figure 8.

" β arr1/2" refers to " β arr1 and β arr2" indeed indicating that both β arr were co-transfected; " β arr1/2 KO" means cells in which both β arr were genetically deleted". We clarified this in the different figure legends.

- Legend figure 3D: could you indicate which Galpha subunits are missing?

The following G proteins are depleted from the G α KO cell line (Δ GNAS/ GNAL/ GNAQ/ GNA11/ GNA12/ GNA13/ GNAI1/ GNAI2/ GNAI3/ GNAO1/ GNAZ/ GNAT1/ GNAT2).

This was added to the text (lines 211-212) and to the figure legend.

- Fig. 8C: why is an excess of V2Rpp necessary?

The point of this experiment was to estimate the affinity between G $\beta\gamma$ and β arr. We added an excess of V2Rpp to make sure that all β arr would be engaged by V2Rpp so that we do not have two populations of β arr that could influence the measured K_d. This has now been clarified on line 278-280.

Reviewer #2 (Remarks to the Author):

The manuscript by Sokrat et al describes the formation and role of β arr and $G\beta\gamma$ in sustained GPCR signalling. The manuscript is well written with clear description of the methods and results as well as relevant discussion. The manuscript uses a range of techniques to demonstrate the formation, trafficking and regulation of β arr- $G\beta\gamma$ complexes. While the functional consequence of formation of these complexes is not investigated in the present manuscript. The discussion on this matter is logical and with only minimal speculation. Moreover, this manuscript provides key mechanistic insight to our understanding of sustained GPCR signalling from endosomes.

I have the following comments on the content of the manuscript for the authors to consider.

Major comments:

I have major concerns regarding the statistical analyses used throughout the manuscript both in terms of detail and tests used. I was unable to find in the methods section details on the statistical tests used nor any justification for their use.

- a. The figure legends do contain the information that the p-values stated are “(unpaired t-test)” although this appears to be a boiler plate sentence and is sometimes included even where no statistical analysis is indicated on the graph e.g. SuppFig3. It would be useful for interpretation of the data for the tests used, with justification, to be outlined in the manuscript.
- b. The authors should justify their use of an unpaired t-test without correction for multiple comparisons throughout the manuscript. Indeed, for many analyses/graphs an ANOVA appears to be more appropriate.

We apologize for the oversight concerning the statistical analyses and appreciate the reviewer's insights on the type of statistical tests to be used. The reviewer is right that in many cases multiple comparisons were involved. We have therefore reanalyzed our data using ANOVA and multiple comparison tests where appropriate and indicated the type of test used in each figure legend. We have also added a “Statistical analysis” section to the Methods in which we describe the statistical test used. It should be noted that the new statistical analysis did not modify the interpretation of the data or the conclusions drawn.

c. From viewing the graphs, it is sometimes unclear why some comparisons were made and another not. For example, Fig2a it appears statistical analysis was performed on the Mock Barr1/2 KO group but not the mock parental group, yet a reduction was also seen in the parental group? In Figure 3B comparisons are made in the text between the vehicle and AVP-induced responses “In parental HEK293T cells (Fig. 3B), expression of β arr2-RlucII with V2R-mTFP (D + I) resulted in an AVP-induced increase in signal, indicative of recruitment of β arr2 to the receptor. Expression of β arr2-RlucII with $G\gamma$ 2-YFP (D + A) did not result in an agonist-induced response in the absence of overexpressed V2R,…” yet there are no comparisons made on the graph.

d. Some graphs indicate significance with asterisk that are not linked to another group making assessment of the comparison impossible without further detail in the figure or graph. Again, notable examples are Fig2a but also Fig 2b, Figure 6d, SuppFig 2. Which groups are being compared should be made clear either on the graph or figure legend.

We have modified our representation of the statistical comparisons to better illustrate which conditions are being compared and added statistical analysis to conditions that were compared in the text and not in the figures.

Minor comments

1 pg6: The authors state that “release from the plasma membrane was found to be restricted to this compartment since anchoring β ARKct to the endosomes using the FYVE targeting domain of endofin” Yet in Figure 1b FYVE targeted β ARKct appears to significantly reduce the change in BRET in the presence of additional $\beta\gamma$. Can the authors comment on a mechanism of action for this observation.

This is an interesting point raised by the reviewer. One hypothesis we can propose is that the potentiating effect of overexpressing $G\beta\gamma$ on $G\alpha$ dissociation from the plasma membrane is attenuated by overexpression of β ARKct in the FYVE containing domain (endosomes) as a consequence of scavenging $G\beta\gamma$ in this compartment, thus reducing its impact at the plasma membrane. This was addressed in the results section (line 122-125).

2. pg10, The authors present data on the role of $G\alpha$ subunits in the formation of $V2R-\beta$ arr- $G\beta\gamma$ complexes (Fig6). They also note that such complexes can exist “as a unique entity without the incorporation of $G\alpha$ subunits.” based on the data in Fig 3. While not contradictory there does appear to be a decrease in the BRETfect ratio between the Fig3B and Fig3D for the AVP induced response potentially further supporting the authors observations in Fig 6. The authors may wish to comment on this.

The reviewer is right, the data from figures 3, 5 and 6 are consistent; while the $V2R-\beta$ arr- $G\beta\gamma$ complex can form in total $G\alpha$ KO cells, total BRETfect signal as well as complex formation kinetics are decreased. Our explanation is that since $G\alpha$ s couples directly to activated $V2R$, the presence of a functional heterotrimer delivers $G\beta\gamma$ to $V2R$ - β arr complex more efficiently. We have expanded upon our explanation regarding that matter in our discussion (line 314-323).

3. Supplementary Figure 1: It is not clear from the methods or figure legend the difference between D+I and D+I+A in S1 and Fig3B? As interpreted the figure legend indicates that only D+A has the addition of $V2R$. Are these the same data or additional replicates? If additional replicates without the addition of unlabelled $V2R$ it is unclear what the addition of the D+I and D+I+A groups adds to the manuscript. There are also no p-values indicated on the figure.

In the figure 3B, there is no agonist-induced increase in signal in the D+A condition, most likely due to a low level of expression of the $V2R$. The data shown in the original Fig S1 (now Fig S4) represent additional experiments where we overexpressed unlabeled $V2R$ with β arr2-RlucII (D) and $G\gamma_2$ -YFP (A) to show that the presence of $V2R$ enables agonist-induced β arr2- $G\beta\gamma$ interaction. The D+I and D+I+A groups are additional replicates added to show that we could see association of β arr and $G\beta\gamma$ with unlabeled $V2R$ but that the BRETfect signal does not reach the level of the D+I+A condition where the $V2R$ -mTFP potentiates the increase in signal. We added the statistical analysis that was missing to this figure.

Reviewer #3 (Remarks to the Author):

The manuscript by Badr Sokrat#, Anthony H. Nguyen#, et al. entitled

“The V2R– β arrestin–G $\beta\gamma$ complex promotes G protein translocation to endosomes”

Describes the dissociation of the G α subunit from the active G-protein at the plasma membrane and obvious re-association with the receptor on endosomes. This is a nice and timely study and well written with easy to follow arguments backed by several well thought control experiments.

Introduction:

- Clearly written! Therefore, I only have very minor points

- Line 57/58: I would suggest to use “expected of a receptor-activated β arr” instead of “expected of an activated β arr” to be more specific since there may be other active states of arrestin e.g. binding to JNK3 that favor a different state of arrestin but this arrestin is still active....

Thank you for the suggestion. This was modified in the manuscript.

- Line 64-67: “Several class B GPCRs such as the parathyroid hormone receptor (PTHrP), neurokinin 1 receptor (NK1R) and the vasopressin type 2 receptor (V2R) have been shown to continue signaling within internalized compartments instead of remaining desensitized.” Please indicate the appropriate references for these receptors and finding.

Thank you for the suggestion. The appropriate references were added to the introduction.

Results section:

- “G α s dissociates from the plasma membrane after V2R activation and translocates to endosomes”:

o Could the authors please reason why they used the time-point of 20 min to look at AVP-induced G α s dissociation from the plasma membrane? Is this a value taken from literature? Did the authors check the signal over time and then decided on this time point as it reaches a stable plateau? In that case it would be great to see the measurement over time in the supplements. This would be nice/helpful to know for the readers.

We indeed conducted kinetics of G α trafficking from the plasma membrane to the endosomes and found that at 20 minutes the BRET window was optimal for both plasma membrane and endosomal sensors. We chose to measure both G α dissociation from the plasma membrane and trafficking to the endosomes at same time point even though G α dissociation can reach its maximal more rapidly after stimulation. The kinetics of G α trafficking were added to the manuscript as the new Fig. Supplementary 1 and are now referred to in the results section (lines 103 – 109).

o Please show that the G α and G $\beta\gamma$ actually dissociate to make clear that it is really reconstitution of the heterotrimers that the authors measure in this assay! (Also would add more confidence to lines 139-141 “Endosomal G $\beta\gamma$ could then attract the G α released from the plasma membrane, allowing the reconstitution of a trimeric G protein in the endosomal compartment.”)

To address the reviewer’s point, we performed two assays to show G α and G $\beta\gamma$ dissociation. First, we measured the agonist-induced BRET decrease between RlucII-117-Gas and GFP10-G γ 2 indicating dissociation of the G α and G $\beta\gamma$ subunits. In addition, we also showed recruitment of GRK2 by the G $\beta\gamma$ dimer that requires dissociation of G α (Koch WJ. et al., JBC, 1993; Lodowski, DT., et al., Science, 2003). These two assays clearly show dissociation of G α and G $\beta\gamma$ and were added to the manuscript as new Supplementary Figure 2 and are now discussed in the results section (lines 145-148).

**RlucII-117Gas / GFP10-Gy2
FLAG-V2R**

**Gy2-RlucII / GRK2-GFP10
FLAG-V2R**

o Does the overexpression of the full GRK2 lead to the same effect? This would be an interesting add on to contextualize the finding and show how the full GRK2 protein influences this mechanism as it has been shown that GRK2 expression varies in different cell types but also in pathophysiological conditions!

To address the question raised by the reviewer, we tested the effect of overexpressing full length GRK2 on the trafficking of *Gas*. As you can observe below, GRK2 has similar effects compared to the expression of β ARKct by reducing the trafficking of *Gas* from the plasma membrane to the endosomes. This suggest that the full length GRK2 is able to scavenge $G\beta\gamma$, which is not surprising. However, considering that overexpression of GRK2 also has a significant impact on the phosphorylation and subsequent internalization of GPCRs, it is difficult to determine precisely the mechanism explaining this effect. Not to create confusion we elected not to add this data to the manuscript but could do it if the reviewer found it essential. We believe that this question deserves its own line of investigation and hope to pursue this in the future.

**Gas67-RlucII / rGFP-CAAX
FLAG-V2R**

**Gas67-RlucII / rGFP-FYVE
FLAG-V2R**

o Lines 112-114: “These results suggest that the formation of heterotrimeric Gs and dissociation of G α s from the plasma membrane is dependent on the presence of free G $\beta\gamma$.” As the formation of heterotrimeric Gs is not directly measured here, shouldn’t the phrasing of the causality rather be “These results suggest that the dissociation of G α s from the plasma membrane is dependent on the presence of free G $\beta\gamma$ and the formation of heterotrimeric Gs.”? Because the presence of the inhibitory G $\beta\gamma$ scavenger β ARKct-CAAX reduces and the increased availability of free G $\beta\gamma$ by overexpression drastically enhances the dissociation of G α s from the plasma membrane, hence the interaction with G $\beta\gamma$ and the formation of a heterotrimer is probably necessary for this process.

We thank the reviewer for this comment. This was modified in the manuscript (lines 116-118).

o Lines 114- 118: “The effect of scavenging free G $\beta\gamma$ with β ARKct on G α s release from the plasma membrane was found to be restricted to this compartment since anchoring β ARKct to the endosomes using the FYVE targeting domain of endofin44 had little impact on G α s dissociation from the plasma membrane with or without G $\beta\gamma$ overexpression (Fig.1B).” The dissociation under conditions when G $\beta\gamma$ is overexpressed is significantly reduced. How would the authors interpret that?

This question was also raised by reviewer 2 and the answer that we provided is copied here: This is an interesting point raised by the reviewer. One hypothesis we can propose is that the potentiating effect of overexpressing G $\beta\gamma$ on G α dissociation from the plasma membrane is attenuated by overexpression of β ARKct in the FYVE containing domain (endosomes) as a consequence of scavenging G $\beta\gamma$ in this compartment. Thus, reducing its impact at the plasma membrane. This is now addressed in the results section (line 122-125).

o In line lines 120, 121: the authors write “suggesting that β arr is not required for G α s dissociation from the plasma membrane (Fig. 1A).”; lines 129-131 “Strikingly, we observed a significant reduction in G α s translocation to endosomes in β arr1/2 KO cells (Fig. 1C), whereas G α s dissociation from the plasma membrane was not impaired by β arr1/2 depletion (Fig. 1A).”; lines 133, 134: “However, depletion of β arr only impairs G α s endosomal translocation, not its

dissociation from the plasma membrane.”

This is quite confusing since the reduction of BRET signal in the CAAX assay might not actually reflect a dissociation from the plasma membrane but rather re-distribution processes within the membrane that are happening independently from β arr1/2 while the actual dissociation from the plasma membrane and trafficking to early endosomes is significantly dependent on β arr1/2.

Please consider re-phrasing this section a bit or comment on where the authors would suggest the $G_{\alpha s}$ goes if it does not reach the early endosomal stage in β arr1/2 KO cells?

Numerous studies have shown that $G_{\alpha s}$ dissociates from the plasma membrane and goes to the cytoplasm after receptor activation (Wedegaertner, P. B. et al., Cell, 1994; Yu, J. Z. et al., Mol Pharmacol, 2002). In addition, a recent study (Thomsen, A. R. B. et al., Cell, 2016, Fig. S3, see below) showed that at basal level, $G_{\alpha s}$ is located at the plasma membrane while β arr is mainly cytoplasmic. Receptor activation results in rapid recruitment of β arr to the plasma membrane and simultaneous translocation of $G_{\alpha s}$ to the cytoplasm. Interestingly, there is no colocalization between $G_{\alpha s}$ and β arr prior to their translocation to the endosomes indicating that $G_{\alpha s}$ does not traffic with the receptor- β arr complex and could be independent of β arr. Yet, the reviewer is right that we cannot exclude that the BRET decrease between $G_{\alpha s}$ -RlucII and rGFP-CAAX could result in part from a redistribution in the membrane and not only from its dissociation from the plasma membrane. A statement to this effect has been added to the discussion (lines 303 and 310-313).

Fig S3: Cellular Localization of SNAP-V₂R Pre-labeled with SNAP-Surface 649 Fluorescent Substrate, mStrawberry-βarr2, and mEmerald-67-Gas Visualized by Confocal Microscopy (Thomsen, A. R. B. et al., Cell, 2016).

- “βarr mediates Gβγ trafficking from the plasma membrane to the endosomes”:

o Please include a measurement of V₂R-RlucII and CXCR4-Rluc to rGFP-FYVE to show that the receptors reach the endosomes and that this is not just an observed process in the membrane.

To address this point, we have measured the BRET signal of V₂R-RlucII and CXCR4-RlucII with rGFP-FYVE. As shown in Figure Supplementary 3, in agreement with what was observed with the rGFP-CAAX sensor, that the trafficking of the V₂R to the endosome is dependent on βarr while CXCR4 can traffic from the plasma membrane to the endosomes in the absence of βarr. These results were added to the manuscript as a new Supplementary Figure S3 and referred to in the text (lines 171-174).

o “In contrast to what is observed for the V2R, for which the loss of plasma membrane receptor upon activation requires β arr (Fig. 2A).” There still seems to be some residual signal left (around 10%) in β arr-KO cells, even if the amplitude is very much reduced compared to the CXCR4. Could the author explain what this process is? 10% signal with such small error bars is not nothing....

The reviewer is right that there seem to be some residual endocytosis occurring in the β arr KO cells even for the V2R. This most likely indicate the contribution of an alternative mechanism that does not rely on β arrestin. This is the object of significant investigations for different GPCR. We acknowledge this point in the results section (lines 164-166).

- “V2R, β arr and $G\beta\gamma$ form a complex in cells”

o The authors state in lines 197, 198 “As seen in Fig. 3D, V2R- β arr2- $G\beta\gamma$ complex could still form in the total absence of $G\alpha$ proteins.” Could the authors please comment/explain the loss of measured BRETfect signal amplitude in total $G\alpha$ KO cells? Could it be that the presence of $G\alpha$ somehow enhances the effect/makes the process more efficient? à lines 213-215 “Although the formation of the V2R- β arr2- $G\beta\gamma$ complex was also observed in the total $G\alpha$ KO, the formation kinetics was much slower ($t_{1/2}$: 288.8 sec) in the absence of $G\alpha$ subunits (Fig. 5B-C).” This fits quite well with the observation mentioned above for Fig. 3D?! Please comment.

This question was also raised by reviewer 2 and the answer that we provided is copied here:

The reviewer is right, the data from figures 3, 5 and 6 are consistent; while the V2R- β arr- $G\beta\gamma$ complex can form in total $G\alpha$ KO cells, total BRETfect signal as well as complex formation

kinetics are decreased. Our explanation is that since $G_{\alpha s}$ couples directly to activated V2R, the presence of a functional heterotrimer delivers $G\beta\gamma$ to V2R- β arr complex more efficiently. We have now further our explanation in that respect in the discussion (line 314-323).

- “Agonist-promoted V2R- β arr2- $G\beta\gamma$ complex formation occurs at the plasma membrane”
 - o Lines 225, 226: “In contrast, over-expression of $G_{\alpha i}$ and $G_{\alpha 12}$, blunted complex formation (Fig. 6B).” “Blunted” seems like a strong word here. The signal reduction is visible and the signal is clearly diminished but blunted.

We changed “Blunted” to “reduced”.

- Figure 1:

- o Which AVP concentration was used to stimulate the receptor? Please definitely include this information in the figure legend and potentially also method section!

We added this information (100nM AVP) to the different figure legends.

- o In panel C the lower labeling seems to be cut off, please make sure that this is not the case in the final figure so that every label is legible.

We modified the figure to fix this issue.

- o Might be helpful to include panel headings to make it easier to follow for the readers and make directly clear what they are looking at. E.g. “ $G_{\alpha s}$ dissociation from PM” for panel A

We added titles on the side of Figure 1 to make it easier to follow what proteins and cellular compartments are tested in each panel.

- o Legend for A): “Overexpression of β ARKct-CAAX and $G\beta 1\gamma 2$ modulates $G_{\alpha s}$ dissociation from the plasma membrane.” Instead of “Overexpression of β ARKct-CAAX and $G\beta 1\gamma 2$ modulates $G_{\alpha s}$ dissociation.” to be more specific.

We corrected this in the Figure 1 legend.

- o Panel B, D: Please label “ $\beta\gamma$ ” as “ $G\beta\gamma$ ” as in the rest of the figure and manuscript to be consistent.

We corrected this in Figure 1.

- Figure 4:

o I am aware that BRET microscopy is a great achievement but could the resolution of the images be improved? (Also for images in Fig.5)

As indicated by the reviewer, getting nice BRET images is very challenging. These are the first BREFTect images published and although the resolution is not perfect, we believe that the discussed phenomenon is clearly seen especially in the videos included in the manuscript. Yet, we added new images to Figure 5 that show improved resolution and clearer increase in signal at the plasma membrane.

o Could the authors provide some kind of quantification that is the underlying basis for the coloring? Maybe as a supplementary table?

We added a quantitative scale to the microscopy figures. The numeric scale of the heat-map legend represents calculated BRET ratios.

- Figure 5:

o I don't want to be picky but to put the signal scale in the middle of the figure and have time traces follow with the same colors is more than misleading. I would suggest to put the scale bar to the right of the cell pictures.

We apologize that this may have appeared misleading. It was certainly not our intention. We modified the colors in Figure 5C and moved the scale to the right of the microscopy images.

- Figure 9:

o Why do the authors indicate GRK2 specifically in this overview? The authors did not test for this. What do the authors refer to? Do they want to indicate that there are isoform-specific differences? Otherwise the authors could just leave out the specification of the number 2.

This is a good point by the reviewer, we mistakenly indicated GRK2 where it could be any other GRK. We corrected this in Figure 9.

Discussion:

- There is indication for bArr1 interaction with GNB1 in the literature, roughly localized to amino acids 180-280 in bArr1, as listed by Crepieux et al 2017 and found in Yang, 2009, Biochem. J., 417, 287. This information could be added for the reader.

We are now citing the findings of Yang, et al. 2009 and Crepieux, et al. 2017 regarding β arr1 interaction with G β 1 (line 78-79).

- Generally, it would be very helpful to follow the argumentation of the authors if they could include the appropriate reference to their results figure in the statements they make in the discussion about their results, so that it is easier for the readers to follow the line of thought.

We added references to the relevant results in the discussion to facilitate the reading.

Methods:

- I know this Gas67 has been around for quite some time but could the authors mention for the younger researchers what exactly the Gas67 construct is and what the 67 means, so that it is clear what kind of constructs were used in this study (Older work still deserves to be cited).

We explained how the G α s67-RlucII construct was generated and added citations of previous work that developed this tool in the methods section (line 377).

REVIEWERS' COMMENTS:

Reviewer #1 (Remarks to the Author):

The authors have answered all my questions.

Reviewer #2 (Remarks to the Author):

I thank the authors of Sokrat et al for addressing my previous comments. These have answered my original concerns and I do not have any further suggestions arising from the additions made in response to the other reviewer's comments. I believe the manuscript presents interesting results which should be of keen interest to the field.

However, I have two very minor comments

On pg 7 ln 134-136 the authors state: "ARKct-CAAX as well as BARKct-FYVE completely blocked Gas trafficking to the endosomes while overexpression of GBy significantly enhanced Gas endosomal translocation (Fig. 1C-D)." While Fig1D indicates there is statistically significance for enhanced translation by overexpression of GBy, there is none Fig1C where overexpression is "NS". On the balance of the data, I'd agree with the authors that there is enhanced translocation by overexpression of GBy with the analysis in Fig1C perhaps a statistical quirk due to multiple comparisons.

Nevertheless, could the text be updated to reflect that the 'significant' enhancement is only observed in Fig1D.

In figure 2 panels A and B are labelled as "Agonist-induced receptor internalisation" are these change in BRET or a % change – for clarity could these be updated to reflect the units.

Reviewer #3 (Remarks to the Author):

The authors have done a great job to answer all my comments and addressed them completely.

I have no more criticism.

REVIEWERS' COMMENTS:

Reviewer #1 (Remarks to the Author):

The authors have answered all my questions.

Reviewer #2 (Remarks to the Author):

I thank the authors of Sokrat et al for addressing my previous comments. These have answered my original concerns and I do not have any further suggestions arising from the additions made in response to the other reviewer's comments. I believe the manuscript presents interesting results which should be of keen interest to the field.

However, I have two very minor comments

On pg 7 In 134-136 the authors state: "BARKct-CAAX as well as BARKct-FYVE completely blocked $G\alpha_s$ trafficking to the endosomes while overexpression of $G\beta\gamma$ significantly enhanced $G\alpha_s$ endosomal translocation (Fig. 1C-D)." While Fig1D indicates there is statistically significance for enhanced translocation by overexpression of $G\beta\gamma$, there is none Fig1C where overexpression is "NS". On the balance of the data, I'd agree with the authors that there is enhanced translocation by overexpression of $G\beta\gamma$ with the analysis in Fig1C perhaps a statistical quirk due to multiple comparisons. Nevertheless, could the text be updated to reflect that the 'significant' enhancement is only observed in Fig1D.

The text has been changed to:

β ARKct-CAAX as well as β ARKct-FYVE completely blocked $G\alpha_s$ trafficking to the endosomes while overexpression of $G\beta\gamma$ enhanced $G\alpha_s$ endosomal translocation found to be significant in Fig. 1d and to be a tendency in Fig. 1c, in agreement with the statistically significant increase in dissociation from the plasma membrane (Fig. 1a).

In figure 2 panels A and B are labelled as "Agonist-induced receptor internalisation" are these change in BRET or a % change – for clarity could these be updated to reflect the units.

(% of basal V2R-RlucII/rGFP-CAAX signal) has been added to the legend of the y-axis of Fig. 2a and b.

Reviewer #3 (Remarks to the Author):

The authors have done a great job to answer all my comments and addressed them completely. I have no more criticism.